# Extended Flow Matching :
# A Method of Conditional Generation with Generalized Continuity Equation

## Abstract

Conditional generative modeling (CGM), which approximates the conditional probability distribution of data given a condition, holds significant promise for generating new data across diverse representations. While CGM is crucial for generating images, video, and text, its application to scientific computing, such as molecular generation and physical simulations, is also highly anticipated. A key challenge in applying CGM to scientific fields is the sparseness of available data conditions, which requires extrapolation beyond observed conditions. This paper proposes the Extended Flow Matching (EFM) framework to address this challenge. EFM achieves smooth transitions in distributions when departing from observed conditions, avoiding the unfavorable changes seen in existing flow matching (FM) methods. By introducing a flow with respect to the conditional axis, EFM ensures that the conditional distribution changes gradually with the condition. Specifically, we apply an extended Monge–Kantorovich theory to conditional generative models, creating a framework for learning matrix fields in a generalized continuity equation instead of vector fields. Furthermore, by combining the concept of Dirichlet energy on Wasserstein spaces with Multi-Marginal Optimal Transport (MMOT), we derive an algorithm called MMOT-EFM. This algorithm controls the rate of change of the generated conditional distribution. Our proposed method outperforms existing methods in molecular generation tasks where conditions are sparsely observed.

## 1 Introduction

*Conditional generative modeling (CGM)*, which involves approximating a conditional probability distribution $p(x \mid c)$ of data $x$ given condition $c$, holds great promise for generating new, previously non-existent data across a wide range of representations. Currently, CGM is pivotal in generating images, videos (Rombach et al., 2021; Saharia et al., 2022a;b; Voleti, 2023), and text (Li et al., 2022; Strudel et al., 2022; Gao et al., 2024), but it is also expected to be applied to scientific computing, such as molecular generation (Kang & Cho, 2019) and physical simulations (Huang et al., 2024; Gebhard et al., 2023).

One of the key challenges of applying CGM in scientific fields is the sparsity of available data conditions. This sparsity necessitates extrapolating beyond the observed conditions (Lee et al., 2023). An important example of scientific applications is molecular generation—imagine that you wish to discover a new molecule $x_{\text{desired}}$ with a desired chemical property $c_{\text{desired}}$, for which no molecular data may be available. Here, we have only observed a limited number of properties $c_{\text{obs}}$, which may be very sparse and require difficult extrapolation. This sparsity issue is more apparent when the condition or property is multi-dimensional.

In contrast, recent deep generative models for CGM have been designed mainly for situations where the conditions are densely observed. Consider the example of methods (Ding et al., 2021; Zhao et al., 2024; Ding et al., 2024) based on Vicinal risk minimization (VRM) by Chapelle et al. (2000). In VRM, the observed conditions $c_{\text{obs}}$ are augmented with Gaussian noise $w_c \sim \mathcal{N}(0, I)$, and the generative model is trained so that the unknown conditional distribution $p(x \mid c_{\text{obs}} + w_c)$ becomes close to the known distribution $p(x \mid c_{\text{obs}})$. Thus, if we can only observe two conditions $c_{\text{obs}}^1$ and

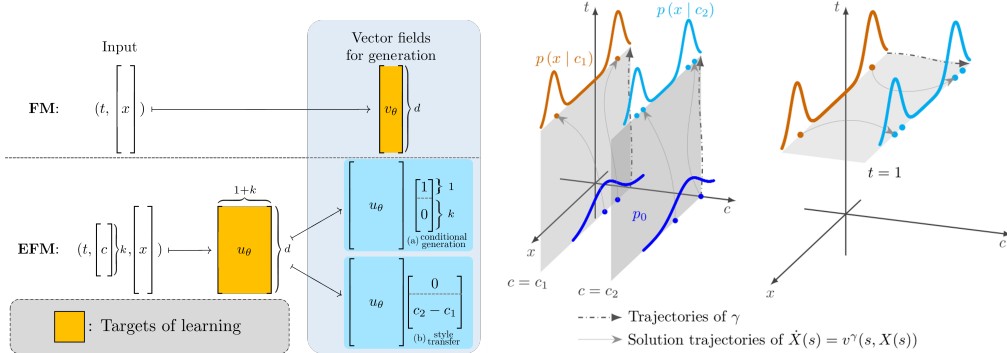

Figure 1: Difference between FM and EFM.

Figure 2: Visualization of the flow for (a) conditional generation along $\gamma^{c_1}$ and $\gamma^{c_2}$ (Algorithm 2), and (b) style transfer along $\gamma^{c_1 \to c_2}$ (Algorithm 3).

$c^2_{\text{obs}}$, which are somewhat distant from each other, then we cannot introduce any inductive bias into the interpolated or extrapolated condition $c_{\text{desired}}$. As a result, the accuracy of the generation of data given $c_{\text{desired}}$ would not improve. Indeed, Figure 4b will show another example where the quality of the generation at $c = c_{\text{desired}}$ deteriorates compared to $c = c_{\text{obs}}$ if no bias is introduced.

We expect that one of the hopes to overcome this difficulty is dynamical generative models, including diffusion models (Song et al., 2021; Ho et al., 2020) and, in particular, the simplest of these—— Flow matching (FM) (Liu et al., 2023; Lipman et al., 2023; Albergo & Vanden-Eijnden, 2023). FM itself is the method of generative modeling to approximate a probability distribution $p(x)$. In FM, two probability distributions are *gradually* deformed by flows induced by ordinary differential equations (ODEs). This deformation makes it possible to formulate the learning of the generative model as an estimation of the "vector field", i.e., the way in which the ODE infinitesimally transformed the data. In particular, the methods based on FM stabilize the learning of vector fields, making it possible to generate a variety of data representations, including images (Esser et al., 2024), text (Hu et al., 2024), audio (Le et al., 2023), DNA (Stark et al., 2024), and molecules (Song et al., 2023; Miller et al., 2024).

This paper proposes the framework of *Extended Flow Matching (EFM)*, which realizes a "smooth" change of distributions for departure from the observed conditions, where we introduce an inductive bias of low sensitivity of $p(x \mid c)$ with respect to conditions $c$. If we assume that the target data is in nature, such as molecules, it is reasonable to impose this inductive bias. We remark that this kind of inductive bias has been used throughout the history of generative models as a method to prevent overfitting and a method to stabilize generative models; see, e.g., (Miyato et al., 2018). Therefore, our method addresses extrapolation by learning a model such that the data to be extrapolated follows this inductive bias of low sensitivity.

More specifically, we apply the extended Monge–Kantorovich theory introduced by Brenier (2003) to conditional generative models. This leads to a framework for learning *matrix fields* in a generalized continuity equation instead of vector fields in the continuity equations in FM.

Furthermore, by combining the concept of Dirichlet energy on Wasserstein spaces introduced by Lavenant (2019) with Multi-Marginal Optimal Transport (MMOT), we can derive an algorithm called *MMOT-EFM* that reduces the sensitivity of the generated conditional distribution. In addition, our proposed method is shown to outperform existing methods in the task of molecular generation in situations where conditions are sparsely observed.

NOTATION

Let us use $\cdot$ to denote a placeholder, $\|\cdot\|$ to denote the Euclidean norm, and $0_k := (0, \ldots, 0)^\top \in \mathbb{R}^k$ to denote the zero vector. We denote by $\mathcal{P}(M)$ the space of probability distributions on a metric space $M$, and denote by $\delta_x \in \mathcal{P}(M)$ the delta distribution supported on $x \in M$. For a distribution

$\mu \in \mathcal{P}(M)$ on $M$ and a vector-valued function $f$ on $M$, we denote by $\mathbb{E}_{X \sim \mu}[f(X)]$ the expectation of a random variable $f(X)$, where $X \sim \mu$ is a random variable following $\mu$.

We also denote $I := [0, 1]$ and $[m : n] := \{m, m + 1, \ldots, n\}$ for $m, n \in \mathbb{N}$ such that $m < n$. For a function $g$ on $I$, we write $\dot{g}(t)$ for the derivative $\frac{\mathrm{d}g}{\mathrm{d}t}(t)$ with respect to time $t \in I$. Further, we let $D \subset \mathbb{R}^d$ be the data space. For any subscript $\xi$, we will denote by $p_\xi$ the density of a probability distribution $\mu_\xi$ on $D \subset \mathbb{R}^d$, i.e., $\mu_\xi(\mathrm{d}x) = p_\xi(x)\mathrm{d}x$ in a measure-theoretic notation. In the following mathematical discussion, we will assume that any probability distribution has a density, but this assumption is superficial and is used only for simplicity of explanation.

## 2 PRELIMINARIES

To motivate EFM, we first present Flow Matching by Lipman et al. (2023) and its variant, OT-CFM (Pooladian et al., 2023; Tong et al., 2023b), through the lens of Monge–Kantorovich theory.

### 2.1 FLOW MATCHING (FM)

**Continuity Equation:** As a method of generative modeling, the goal of FM is to learn a map that transforms a source distribution to a target distribution in the form of $\mu : [0, 1] \to \mathcal{P}(D)$, where $D$ is the space of dataset. Instead of learning $\mu$ directly, flow matching as a method learns a vector field $v : [0, 1] \times D \to \mathbb{R}^d$ such that the *continuity equation* (CE)

$$\partial_t p_t(x) + \mathrm{div}_x(p_t(x)v(t, x)) = 0 \quad ((t, x) \in [0, 1] \times D) \tag{2.1}$$

holds with respect to the density $p_t$ of $\mu_t$, and we use this $v$ for the sample generation.

**Inference:** $X_1 \sim \mu_1$ can be sampled by solving the ODE with $\dot{X}(t) = v(t, X(t))$, $X(0) \sim p_0$.

### 2.2 OT-CFM

OT-CFM, which has been proposed to use optimal transport for constructing the vector field, can be interpreted as a method of minimizing the Dirichlet energy, or the energy of transport for $\mu$ conditional to the boundary condition $\mu_0 = \mu_{\mathrm{source}}, \mu_1 = \mu_{\mathrm{target}}$. Specifically, we will show that a straight line in the construction of OT-CFM can be regarded as a minimizer of the Dirichlet energy.

**Objective energy:** Formerly, Dirichlet or the kinetic energy of the curve $\mu$ can be written as

$$\mathrm{Dir}(\mu) := \inf_{v : I \times D \to \mathbb{R}^d} \left\{ \frac{1}{2} \iint_{I \times D} \|v(t, x)\|^2 p_t(x)\mathrm{d}x\mathrm{d}t \ \middle| \ \text{The pair } (p, v) \text{ satisfies (2.1)} \right\}. \tag{2.2}$$

**Objective function:** To derive the algorithm used in OT-CFM, we first introduce some definitions. Let $Q$ be a distribution over a space $H(I; D) := \{\psi : I \to D \mid \psi \text{ is differentiable}\}$ of paths that map time $t \in I$ to data $x \in D$, $\psi : I \to D$ be a sample from $Q$, and use $\mu_t^\psi$ to denote the delta distribution $\delta_{\psi(t)} \in \mathcal{P}(D)$ supported at $\psi(t) \in D$. With these definitions, we can represent $\mu = \mu^Q$ from $Q$ as

$$\mu^Q : I \ni t \longmapsto \mathbb{E}_{\psi \sim Q}[\mu_t^\psi] \in \mathcal{P}(D). \tag{2.3}$$

As a matter of fact, we can see that the optimal probability path $\mu^{Q^*}$, which minimizes $\inf_Q \mathrm{Dir}(\mu^Q)$ subject to $\mu_0^Q = \mu_{\mathrm{source}}, \mu_1^Q = \mu_{\mathrm{target}}$, is concentrated on the set of "straight lines" $\psi(t \mid x_1, x_2) = tx_2 + (1 - t)x_1$ between joint samples $(x_1, x_2)$ from the target and the source. By (Ambrosio et al., 2008, Theorem 8.2.1), the function $D \times D \ni (x_1, x_2) \mapsto \psi(\cdot \mid x_1, x_2) \in H(I; D)$ allows a parametrization of $Q$ with the optimal transport plan $\pi$ with marginals $\mu_{\mathrm{source}}$ and $\mu_{\mathrm{target}}$. This would allow us to write $\|\psi(t \mid x_1, x_2)\|^2 = \|x_1 - x_2\|^2$ for the optimal $Q^*$. This would reduce the optimization with respect to $Q$ to the classic optimal transport problem for the joint probability $\pi$ with cost $c(x, y) = \|x - y\|^2$. In OT-CFM, this is approximated through batches. Following the same logic as in (Kerrigan et al., 2024a), or our later theorem (Theorem 3.4), the vector field $v$, which generates $\mu^{Q^*}$ via CE can be obtained as the minimizer of

$$\mathbb{E}_{\psi \sim Q^*, t \sim \mathrm{Unif}(I)}[\|v(t, \psi(t)) - \dot{\psi}(t)\|^2] = \mathbb{E}_{(x_1, x_2) \sim \pi^*, t \sim \mathrm{Unif}(I)}[\|v(t, \psi(t)) - \dot{\psi}(t \mid x_1, x_2)\|^2]. \tag{2.4}$$

This derives the learning of $v$ through a neural network $v_\theta$ as shown in Algorithm 5. Indeed, Dirichlet energy that OT-CFM is aiming to minimize is a form of inductive bias regarding the continuity of the *generation* process with respect to time $t$.

In naive application of OT-CFM to conditional generation, $\psi(t)$ is replaced with $\psi(t, c)$ for the target $c$. However the energy of OT-CFM only relates to $\|\partial_t \psi(t, c)\|^2$, unlike our EFM in Section 3.

## 3 THEORY OF EFM

In this section, we extend the standard FM theory to consider conditional probability with conditions $c$ within a bounded domain $\Omega \subset \mathbb{R}^k$. Let $p_c(x) := p(x \mid c)$ be the unknown target conditional probability density, and let $p_{0,c}(x) := p_0(x \mid c)$ be a user-chosen tractable conditional density given $c = (c^i)_{i \in [1:k]} = (c^1, \ldots, c^k) \in \Omega$, such as normal distributions with mean and variance parameterized by $c$. We will use the notation in the previous section, that is, we will denote by $\mu_c$ and $\mu_{0,c}$ the distribution of the probability density function $p_c$ and $p_{0,c}$, respectively.

### 3.1 EXTENSION OF FM

We will present this subsection in parallel with § 2.1.

**Generalized Continuity Equation:** We directly extend the interpretation of FM by extending the domain of $\psi$ in (2.3) from $I$ to $I \times \Omega$, where $\Omega$ is the space of conditions. For brevity, instead of using explicit $I \times \Omega$, we would like to use a general bounded domain $\Xi$ in Euclidean space as an analog of $\Omega$ of the previous section and analogously set the goal of EFM to the learning of $\mu \colon \Xi \to \mathcal{P}(D)$. Now, just like FM, instead of learning $\mu$ directly, EFM aims to learn a *matrix* field $u \colon \Xi \times D \to \mathbb{R}^{d \times \dim \Xi}$ such that *generalized CE* (Brenier, 2003; Lavenant, 2019)

$$\nabla_\xi p_\xi(x) + \operatorname{div}_x(p_\xi(x)u(\xi, x)) = 0 \quad ((\xi, x) \in \Xi \times D) \tag{3.1}$$

holds for the density $p_\xi$ of $\mu_\xi$. Here, div is an extended divergence operator, see Appendix (A.1).

**Inference:** Inference based on the matrix field $u$ is slightly more complicated than in FM, which provides a single vector field to integrate the ODE. Various tasks can be solved solely with the matrix field, including the typical cases of generation and transfer. For $\Xi = I \times \Omega$, the generation given condition $c$ will be performed by transforming $\mu_{0,c} \to \mu_{1,c}$, and the transfer from $c$ to $c'$ by transforming $\mu_{1,c} \to \mu_{1,c'}$. Both are performed by integrating the matrix field along the path in $I \times \Omega$. More precisely, the following result justifies our use of the matrix field $u$ in (3.1) to achieve the goal of conditional generative modeling:

> **Proposition 3.1** (GCE generates $\gamma$-induced CE). *Let $\mu \colon \Xi \to \mathcal{P}(D)$ and $u \colon \Xi \times D \to \mathbb{R}^{d \times \dim \Xi}$ be a probability path and a matrix field, respectively, that satisfy (3.1). Then, for any differentiable path $\gamma \colon I \to \Xi$, the $\gamma$-induced probability path $\mu^\gamma := \mu \circ \gamma$ and the $\gamma$-induced vector field $v^\gamma \colon I \times D \ni (s, x) \mapsto u(\gamma(s), x)\dot{\gamma}(s) \in \mathbb{R}^d$ satisfy the continuity equation, i.e., the density $p^\gamma$ of $\mu^\gamma$ and $v^\gamma$ satisfy $\partial_s p_s^\gamma(x) + \operatorname{div}_x(p_s^\gamma(x)v^\gamma(s, x)) = 0$.*

The rigorous version of Proposition 3.1 is given in Proposition A.2 in the Appendix. Proposition 3.1 shows that the flow on $D$ corresponding to an arbitrary probability path on $\{\mu_\xi \in \mathcal{P}(D) \mid \xi \in \Xi\}$ can be constructed from the $\gamma$-induced vector field obtained from multiplying the matrix $u$ to the vector $\dot{\gamma}$. Thus, once the matrix field $u$ is obtained, the desired vector field $v^\gamma$ is to be calibrated by choosing an appropriate $\gamma$ that suits the purpose of choice. When the pair of $p_\xi$ and $u_\xi$ satisfies GCE (3.1), the designs of $\gamma$ in the following two examples possess significant practical importance (See Figure 1 and Figure 2 ):

*Example* 3.2 (Conditional generation). When the goal is to sample from the unknown conditional distribution $\mu_{c_*}$ given condition $c_* \in \Omega$, we can choose $\gamma^{c_*} \colon I \to I \times \Omega$ such that $\gamma^{c_*}(1) = (1, c_*)$; typically, we can set $\gamma^{c_*}(s) = (s, c_*)$ for $s \in I$. Then, by virtue of Proposition 3.1 and the continuity equation (2.1), we only need to compute the flow $\phi$ by solving the ODE

$$\begin{cases} \dot{\phi}_s(x_0) = u(s, c_*, \phi_s(x_0)) \begin{bmatrix} 1 \\ 0_k \end{bmatrix} (s \in I), \\ x_0 \sim \mu_{0,c_*}, \end{cases}$$

and obtain samples $\phi_1(x_0)$ from $\mu_{1,c_*} = \mu_{c_*}$. The trajectories in the front and rear plane of (a) in Figure 2 respectively represent the flows corresponding to this example with $c_* = c_1$ and $c_* = c_2$.

*Example* 3.3 (Style transfer). When the goal is to transform a sample generated from $\mu_{c_1}$ to a sample of another distribution $\mu_{c_2}$ given $c_2 \in \Omega$, we may choose $\gamma^{c_1 \to c_2} : I \to I \times \Omega$ satisfying $\gamma^{c_1 \to c_2}(0) = (1, c_1)$ and $\gamma^{c_1 \to c_2}(1) = (1, c_2)$. For example, we can set $\gamma^{c_1 \to c_2}(s) = (1, (1-s)c_1 + sc_2)$ for $s \in I$. In this case, we only need to solve the ODE

$$\begin{cases} \dot{\phi}_s(x_0) = u(1, \gamma^{c_1 \to c_2}(s), \phi_s(x_0)) \begin{bmatrix} 0 \\ c_2 - c_1 \end{bmatrix} \ (s \in I), \\ x_0 \sim \mu_{c_1}. \end{cases}$$

The solution trajectories in (b) in Figure 2 represent the flows corresponding to this style transfer.

## 3.2 OBJECTIVE ENERGY AND MMOT-EFM

Now we extend the arguments in § 2.2 to EFM.

**Objective energy:** Just like in § 2.2, we use the representation of $\mu$ as (2.3) through a distribution $Q$ over a space $H(\Xi; D)$ of differentiable maps $\psi$ from $\Xi$ to $D$. Now, the construction of EFM allows us to introduce inductive bias regarding a property of $\psi : \Xi \to D$ and hence how $\mu$ behaves with respect to $\xi$. In particular, if a given energy $\mathcal{E}$ with respect to $\mu^\psi$ is convex, then by Jensen's inequality we can bound $\mathcal{E}(\mu)$ from above by $\mathbb{E}_{\psi \sim Q}[\mathcal{E}(\mu^\psi)]$. Please also see Propositions B.1 and B.2 for more precise statements of these results.

In MMOT-EFM, we consider the case in which $\mathcal{E}$ is the following generalization of the Dirichlet energy (2.2). According to Lavenant (2019), a generalization of Dirichlet energy of a function $\mu : \Xi \to \mathcal{P}(D)$ is given by

$$\text{Dir}(\mu) := \inf_{u : \Xi \times D \to \mathbb{R}^d} \left\{ \frac{1}{2} \iint_{\Xi \times D} \|u(\xi, x)\|^2 p_\xi(x) \mathrm{d}x \mathrm{d}\xi \ \middle| \ \text{The pair } (p, u) \text{ satisfies } (3.1) \right\}, \quad (3.2)$$

where $p_\xi$ is the density of $\mu_\xi$. This energy is of great practical importance because it also measures how large $\mu$ changes with respect to $\xi$.

**Objective function:** Unfortunately, unlike in the case of OT, the energy-minimizing $\mu$ that can be written as $\mu = \mu^Q := \mathbb{E}_{\psi \sim Q}[\mu^\psi]$ is not necessarily achieved with $Q$ concentrated on "straight paths", or (flat) hyperplanes interpolating joint samples from $\{\mu_\xi\}$. Thus we choose to constrain the search of $Q$ to a specific subspace $\mathcal{F}$ of $H(\Xi; D)$, such as Reproducing Kernel Hilbert Space (RKHS). In this search, we also require $Q$ to satisfy the boundary condition (BC) that

$$\mathbb{E}_{\psi \sim Q}\left[\delta_{\psi(\xi)}\right] = \mu_\xi \ (\xi \in A), \quad (3.3)$$

where $A \subset \Xi$ is a finite set for which $\mu_\xi$ ($\xi \in A$) is either known or observed. Instead of (3.3), suppose $\boldsymbol{x}_A := (x_\xi)_{\xi \in A}$ for $A \subset \Xi$ is a joint sample with $x_\xi \sim \mu_\xi$. Then, let $\phi : D^{|A|} \to \mathcal{F}$ be the function-valued mapping, returning the function $\Xi \ni \xi \mapsto \phi(\xi \mid \boldsymbol{x}_A) \in D$ defined by the regression

$$\phi(\cdot \mid \boldsymbol{x}_A) \in \arg\min_{f \in \mathcal{F}} \sum_{\xi \in A} \|f(\xi) - x_\xi\|^2, \quad (3.4)$$

i.e., $\phi(\cdot \mid \boldsymbol{x}_A)$ satisfies $\sum_{\xi \in A} \|\phi(\xi \mid \boldsymbol{x}_A) - x_\xi\|^2 = \min_{f \in \mathcal{F}} \sum_{\xi \in A} \|f(\xi) - x_\xi\|^2$ for each $\boldsymbol{x}_A \in D^{|A|}$. For a joint distribution on $\pi$ on $D^{|A|}$, the parametrization $Q \to \phi_\# \pi$ of random paths allows us to bound the energy from above in the following way:

$$\inf_Q \text{Dir}(\mu^Q) \leq \inf_Q \iint_{H(\Xi; D) \times \Xi} \|\nabla_\xi \psi(\xi)\|^2 Q(\mathrm{d}\psi) \mathrm{d}c \leq \inf_\pi \iint_{D^{|A|} \times \Xi} \|\nabla_\xi \phi(\xi \mid \boldsymbol{x}_A)\|^2 \pi(\mathrm{d}\boldsymbol{x}_A) \mathrm{d}c.$$

Now observe that the upper bound is the form of a marginal optimal transport problem about $\pi$ with marginals $\mu_A$ and $c(\boldsymbol{x}_A) = \int_\Xi \|\nabla_\xi \phi(\xi \mid \boldsymbol{x}_A))\|^2 \mathrm{d}\xi$, whose solution $\pi^*$ can be approximated with batch as in the OT-CFM case. See Table 1 for the parallellism between MMOT-EFM and OT-CFM.

Table 1: Constructions of $\psi\colon [0,1] \to D$ and $\bar\psi\colon \Omega \to D$ and $\pi$ in OT-CFM and MMOT-EFM. Note that they agree when $\mathcal{F}$ is a set of linear functions from $\Omega$ to $D$ and when $\Omega = [0,1] \subset \mathbb{R}$.

| | OT-CFM | MMOT-EFM |
|---|---|---|
| Interpolator | $\psi\left(t \mid x,y\right) = tx + (1-t)y$ | $\bar\psi\left(\cdot \mid \boldsymbol{x} = (x_i)_i\right) \in \underset{\phi\in\mathcal{F}}{\arg\min} \sum_i \|\phi(c_i) - x_i\|^2$ |
| Cost | $\displaystyle \iiint_{[0,1]\times D^2}\|\dot\psi\left(t \mid x,y\right)\|^2 \mathrm{d}t\,\pi(\mathrm{d}x,\mathrm{d}y)$ $\displaystyle (= \iint_{D^2}\|x-y\|^2\pi(\mathrm{d}x,\mathrm{d}y))$ | $\displaystyle \iint_{\Omega\times D^{|C|}} \left\|\nabla_c \bar\psi\left(c \mid \boldsymbol{x}\right)\right\|^2 \mathrm{d}c\,\pi(\mathrm{d}\boldsymbol{x})$ |

Similarly to (2.4), Theorem 3.4 below let us train $u$ corresponding to $\mu^{Q^*}$ via (3.1) as the minimizer of

$$\mathbb{E}_{\psi\sim Q^*,\xi\sim\mathrm{Unif}(\Xi)}[\|u(\xi,\psi(\xi)) - \nabla_\xi\psi(\xi)\|^2] = \mathbb{E}_{\boldsymbol{x}_A\sim\pi^*,\xi\sim\mathrm{Unif}(\Xi)}[\|u(\xi,\psi(\xi)) - \nabla_\xi\phi\left(\xi \mid \boldsymbol{x}_A\right)\|^2] \tag{3.5}$$

which we would use as the objective function of MMOT-EFM. Please also see Lemma A.4.

> **Theorem 3.4.** *Assume we have a random path $\psi \sim Q \in \mathcal{P}(H(\Xi; D))$ that satisfies (3.3) and let $\mu_\xi = \mathbb{E}_{\psi\sim Q}\left[\delta_{\psi(\xi)}\right]$ for $\xi \in \Xi$. For neural networks $u_\theta$, set*
>
> $$\mathcal{L}'(\theta) = \int_\Xi \mathbb{E}_{\psi\sim Q}\left[\|u_\theta(\xi,\psi(\xi)) - \nabla_\xi\psi(\xi)\|^2\right]\mathrm{d}\xi. \tag{3.6}$$
>
> *If there exists a matrix field $u\colon \Xi \times D \to \mathbb{R}^{d\times(1+k)}$ satisfying (3.1), then it follows that $\nabla_\theta\mathcal{L}(\theta) = \nabla_\theta\mathcal{L}'(\theta)$ for $\theta \in \mathbb{R}^p$. Here, we set $\mathcal{L}(\theta) := \int_\Xi \mathbb{E}_{x\sim\mu_\xi}\left[\|(u_\theta - u)(\xi,x)\|^2\right]\mathrm{d}\xi$.*

## 4 TRAINING ALGORITHM

In this section, we leverage the EFM theory of § 3 to construct an algorithm for learning $u_\theta$ in Proposition 3.1, which can be used for conditional generation tasks as well as for style transfer. We summarize the training algorithm in Algorithms 1 and 8.

Because EFM is a direct extension of FM, our algorithm roughly follows the same line of procedures as that of FM (Algorithm 5): (a) sampling data, (b) constructing the supervisory signal $\nabla\psi$, and (c) updating the network by averaged loss. However, in our algorithm, the domain of $\psi$ is $I \times \Omega$ as opposed to just $I$. We developed our algorithm so that, when it is applied to the unconditional case, the trained model agrees with FM. Although the general EFM, as opposed to MMOT-EFM, does not necessarily need to parametrize $Q$ with respect to joint distribution $\pi$, in this paper, we focus on the procedure that uses the joint distribution $\pi$ and $\psi$ in the form of (3.4) and (3.5).

**Step 1 Sampling from Datasets:** Our objective begins from the sampling of $\psi$, whose Jacobian serves as the supervisory signal in the objective (3.5). In order to sample $\psi$, we construct $Q$ from a joint distribution $\pi$ defined over $D^{2N_c}$ with marginals that are approximately $(\mu_{t,c})_{t\in\{0,1\},c\in C_0}$. To this end, we begin by randomly choosing a subset $C_0 := \{c_i\}_{i=1}^{N_c}$ from $C$ so that $C_0$ consists of close points. We then sample a batch $B_{0,c}$ from $\mu_{0,c}$ and $B_{1,c}$ from $D_c$ for each $c \in C_0$. For the reason we describe at the end of this section, we chose $\mu_{0,c} = \mathrm{Law}(R(c) + z)$ with $z$ being a common Gaussian component, and $R\colon \Omega \to D$ is regressed from $\{(c_i, \mathrm{Mean}[D_{c_i}])\}_i$ by a linear map. We choose this option because it theoretically aids us in reducing $\mathrm{Dir}(\mu)$ (See Proposition B.2).

**Step 2 Constructing the supervisory paths:** Given the samples $B = (B_{t,c})_{t\in\{0,1\},c\in C_0}$, we sample $(x_{t,c})_{c\in C_0,t\in\{0,1\}}$ from a joint distribution $\pi$ over $D^{2N_c}$ with support on $B$. In MMOT-EFM, as an internal step, we train the joint distribution $\pi$ with $c(\boldsymbol{x}_A) = \int_{I\times\Omega}\|\nabla_{t,c}\phi\left(t,c \mid \boldsymbol{x}_A\right)\|^2\mathrm{d}t\mathrm{d}c$

---

**Algorithm 1** Algorithm of EFM

---

**Input:** Conditions $C \subset \Omega$, set of datasets $D_c \subset D$ $(c \in C)$, network $u_\theta \colon I \times \Omega \times D \to \mathbb{R}^{d \times (1+k)}$, source distributions $p_0 \left( \cdot \mid c \right) (c \in C)$

**Return:** $\theta \in \mathbb{R}^p$

1: **for** each iteration **do**
   # Step 1: Sample
2:   Sample $C_0$ from $C$, $B_{0,c}$ from $p_0 \left( \cdot \mid c \right)$ and $B_{1,c}$ from $D_c$ $(c \in C_0)$. Put $B^0 \coloneqq \{B_{0,c}\}_{c \in C_0}$, $B^1 \coloneqq \{B_c\}_{c \in C_0}$
     # Step 2: Construct $\psi \colon I \times \Omega \to D$
3:   Construct a transport plan $\pi$ among $B^0$ and $B^1$ #§ 4
4:   Sample $(x_{t,c})_{t,c} \sim \pi$
5:   Define $\psi \colon I \times \Omega \to D$ s.t. (4.1)
6:   Sample $t \sim \mathrm{Unif}(I)$, $c \sim \mathrm{Unif}(\mathrm{Conv}\, C_0)$, where $\mathrm{Conv}\, C_0$ is the convex hull of $C_0$.
7:   Compute
$$\psi_{t,c} \coloneqq \psi(t,c)$$
$$\nabla\psi_{t,c} \coloneqq \nabla_{t,c}\psi(t,c)$$
8:   Update $\theta$ by $\nabla_\theta \|u_\theta(t,c,\psi_{t,c}) - \nabla\psi_{t,c}\|^2$
9: **end for**

---

with $\phi$ solved analytically for (3.4) with $\Xi \coloneqq I \times \Omega$, by e.g., Kernel Regression, Linear regression. When possible, the regression function may be chosen to reflect the prior knowledge of the metrics on $\Omega$ by extending the philosophy of Chen & Lipman (2024) to the space of conditions. In practice, however, the computational cost of MMOT scales exponentially with the number of marginals, so we optimize the joint distributions over $B_1 = (B_{1,c})_{1,c \in C_0}$ only and couple the analogous $B_0$ to $B_1$ via the usual optimal transport. Please see § D.3 for a more detailed sampling procedure. Now, given a joint sample $(x_{t,c})_{c \in C_0, t \in \{0,1\}}$, we construct $\psi$ as

$$\psi \left( t, c \mid x_{0,c}, \boldsymbol{x}_{C_0} \right) = (1-t)x_{0,c} + t\bar{\psi}\left( c \mid \boldsymbol{x}_{C_0} \right) \tag{4.1}$$

where $\bar{\psi}\left( c \mid \boldsymbol{x}_{C_0} \right)$ is the solution of the kernel regression problem for the map $T \colon \mathbb{R}^k \ni c \mapsto x_{1,c} \in \mathbb{R}^d$ with any choice of kernel on $\mathbb{R}^k$. Note that this construction of $\psi$ satisfies the boundary condition (3.3) with $A = \{0,1\} \times C_0$, and generalizes the $\psi$ used in OT-CFM.

**Step 3 Learning the matrix fields:** Thanks to the result of Theorem 3.4, we may train $u_\theta \colon I \times \Omega \to \mathbb{R}^{d \times (1+k)}$ via the loss function being the Monte Carlo approximation of (3.6).

## 5 INFERENCE METHOD

The sampling procedures for style transfer and conditional generation respectively follow Example 3.3 and Example 3.2. For the task of style transfer from $c_0$ to $c_*$, we use the flow along the path $\mu_{1,c_0} \to \mu_{1,c_*}$. For the task of conditional generation with target condition $c_*$, we use the flow along $\mu_{0,c_*} \to \mu_{1,c_*}$. See Algorithms 2 and 3 for the pseudo-codes. When generating a sample for $c^* \notin C$, the source distribution $\mu_{0,c^*}$ is constructed by $R(c^*) + \mathcal{N}(0, I)$ where $R$ is as in training.

## 6 RELATED WORKS

**Guidance-based methods:** Since Lipman et al. (2023), several studies have formalized the use of flow-based models for conditional generation. Some works by (Dao et al., 2023; Zheng et al., 2023) parametrize the vector field $v$ with the conditional value $c$ and guidance scale $\omega \in \mathbb{R}$ as $v(t,c,x) = \omega v_t \left( x \mid \varnothing \right) + (1-\omega)v_t \left( x \mid c \right)$, inspired by the classifier-free guidance scheme of Ho & Salimans (2022). Zheng et al. (2023) showed that if $v_t \left( x \mid c \right)$ approximates the conditional score $\nabla \log p \left( x \mid c \right)$ well, then with the right $\omega$, $v_t(x,c)$ aligns with the sequence of distributions from the standard Gaussian to the target distribution. Hu et al. (2023) created a guidance vector by averaging $v_t(x_{c_{\mathrm{targets}}}) - v_t(x_{c_{\mathrm{others}}})$. However, these methods do not control the continuity of generated $\mu_c$ with respect to $c$, except through the network's architecture. Unlike these, EFM constructs the flow

**Algorithm 2** Generation using the matrix field $u_\theta$

**Input:** Trained $u_\theta$, source distribution $p_{0,0}$, target condition $c_*$
**Return:** A sample $x_1$ from $p\left(\cdot \mid c_*\right)$
  Sample $z$ from source distribution $p_{0,0}$
  Solve the regression problem $R: c \longmapsto \mathrm{Mean}[D_c]$ on $C$
  Set $x_{0,c} = z + R(c)$
  Return $\mathtt{ODEsolve}\left(x_{0,c}, u_\theta(\cdot, c, \cdot)\left[\begin{smallmatrix}1\\0_k\end{smallmatrix}\right]\right)$

**Algorithm 3** Transfer using the matrix field $u_\theta$

**Input:** Trained Network $u_\theta$, source sample $x_0 \sim p_{1,c_1}$ with condition label $c_1$, target condition $c_2$
**Return:** A sample $x_2$ from $p\left(\cdot \mid c_2\right)$
  Return
  $\mathtt{ODEsolve}(x_0, u_\theta(1, \gamma^{c_1 \to c_2}(\cdot), \cdot)\left[\begin{smallmatrix}0\\c_2-c_1\end{smallmatrix}\right])$
  `# `$\gamma^{c_1 \to c_2}$` is defined in`
  `Example 3.3`

for any condition $c \in \Omega$ through the matrix field $u$, which solves GCE, allowing an inductive bias on $\mu_c$'s continuity via the distribution $Q$ of $\psi$. The Dirichlet energy used in EFMcontrols the Lipschitz constant for $\psi$ and $\mu$, ensuring the generation of conditional distributions during training. When $u$ is trained with random conditional paths and appropriate boundary conditions, our EFM theory guarantees that the flow $\phi^{\gamma^c}$ transforms the source to the target conditional distribution whenever $c$ is used in training.

**Dynamical generative models (DGMs) for CGM:** In addition to the VRM-based method mentioned in § 1, there are two other methods: COT-FM (Kerrigan et al., 2024b) and Bayesian-FM (Chemseddine et al., 2024), both based on Conditional Optimal Transport (Hosseini et al., 2024). These methods rely on the relatively weak assumption that the map of conditional distributions $c \mapsto p\left(x \mid c\right)$ is measurable, or can be discontinuous with respect to $c$. In contrast, the learning algorithm of EFM is designed under the assumption that $p\left(x \mid c\right)$ is continuous with respect to $c$. This distinction arises because the former addresses situations where high-dimensional conditions, such as inverse problems of PDEs, can be densely observed, while the latter addresses scenarios where relatively low-dimensional conditions, such as molecular generation, can be sparsely observed. Various other methods for learning CGMs have been proposed, depending on how the data and conditions are available. For example, making the vector field depend on the transport plan $\pi$ (Atanackovic et al., 2024) or obtaining a joint sample $(c, x)$ in a Bayesian manner (Wildberger et al., 2023). Note that these methods are not about continuity with respect to $c$ in the distribution $p\left(x \mid c\right)$.

**Energy principles in DGMs:** We also mention the family of Schrödinger-bridge based methods by (Tong et al., 2023a; Koshizuka & Sato, 2022), which also aims to interpolate between an arbitrary pair of distribution. This family solves the continuity equation while minimizing the regularized energy of the user's choice in the generation process. Kim et al. (2023) also uses Wasserstein Barycenter for distributional interpolation. Multi-marginal stochastic interpolants by Albergo et al. (2024) learn a model that is similar to EFM. The method optimizes not only the vector fields but also the path $\gamma: [0, 1] \to \Omega$ in Proposition 3.1 to minimize kinetic energy. Our MMOT-EFM is novel in that it minimizes the transport cost in a complementary way to the stochastic interpolant. MMOT-EFM trains only a matrix field to minimize Dirichlet energy, which is a generalization of the kinetic cost. This makes it possible to learn a model that transports optimally without optimization of $\gamma$.

## 7 EXPERIMENTS

We conducted experiments to investigate our method in applications.

### 7.1 SYNTHETIC 2D POINT CLOUDS

Figure 3: Train data in § 7.1

We first demonstrate the performance of our method on a conditional distribution consisting of synthetic point clouds in a two-dimensional domain $D \subset \mathbb{R}^2$. Here, we consider the case where the space $\Omega$ of the condition is square, i.e., $\Omega = [0, 1]^2$, and train the model when only samples from the conditional distributions $p\left(\cdot \mid c\right)$ at the four corner points $c$ of the square $\Omega$ can be observed,

see Figure 6 in Appendix. We compared our method against COT-FM (Chemseddine et al., 2024;

(a) Wasserstein distance.

(b) Generated points.

(c) Transfer.

Figure 4: Results of § 7.1. Figures 4b and 4c visualize $\phi_s$ in Examples 3.2 and 3.3, respectively.

Kerrigan et al., 2024b), as well as OT-CFM (Tong et al., 2023b) and GG-EFM with the plan $\pi$, which is constructed in the way of generalized geodesic, see § E.

See Figures 4b and 4c for the generation and transfer visualizations, and see Figure 4a for the error between GT and predicted distributions. Note that our method, MMOT-EFM, performs competitively with all its rivals in interpolation and generation tasks. Also, note that the style transfer with MMOT-EFM preserves the structure of the inner and outer clusters.

## 7.2 MNIST with background

As another proof of concept, we compared EFM against Guided-flows (Zheng et al., 2023) on the colored/rotated MNIST dataset with a background of a CIFAR-10 image. In this experiment, we compress the image into a 16-dimensional latent vector space using a pre-trained Wasserstein autoencoder (WAE) in Tolstikhin et al. (2018). We conditioned each image with the rotation angle and (normalized) RGB color of the digit, constituting four dimensional $c \in [0,1]^4 =: \Omega$, where we also normalize the rotation angle so that $180°$ becomes 1. For training, we used 12 conditions uniformly sampled from $[0,1]^4$. This is a very difficult setting even to exclusively learn the condition of color because 12 uniformly sampled conditions in 4-dimensional space are very sparsely located with no apparent structures like a grid. With the above settings, we evaluated the extra/interpolation performance of the EFM as in § 7.1. On the right

of Figure 5, we plot the error $W_1(\mu_c, \hat{\mu}_c)$ against $d(c, C) := \min_{c' \in C} d(c, c')$ for each grid point $c \in \{(c^i)_{i=1}^4 \in [0, 1]^4 \mid c^i \in \{0, 0.5, 1\} \text{ for } i \in [1:4]\}$. Our model performs competitively in terms of $W_1$ distance for the generation of distributions with arbitrary conditions.

### 7.3 CONDITIONAL MOLECULAR GENERATION

Molecular design applications often require the simultaneous consideration of multiple chemical properties. Most traditional molecular design methods combine all property requirements and their constraints into a single objective function. We applied MMOT-EFM to the task of generating constraints for the following two simultaneous properties of molecules in the ZINC-250k dataset by Gómez-Bombarelli et al. (2018): (1) the number of rotatable bonds and (2) the number of hydrogen bond acceptors (HBAs). The experimental setup is described in detail in § F. We first trained a VAE model to encode molecular structures into a 32-dimensional latent space and then trained EFM to perform out-of-distribution conditional generation over this latent space. We measure the MAE between the condition and actual value of the generated compounds. As shown in Table 2, our method outperforms all baseline methods on the averaged MAE for out-of-distribution conditional generation.

## 8 CONCLUSION

In this paper, we developed the theory of EFM, an extension of FM that models the transformation of distributions with respect to conditions by a matrix field. EFM explicitly shows how distributions change under different conditions. The EFM theory is complementary to many powerful existing ideas, particularly through the design of $\psi$ and $Q$. We also introduce MMOT-EFM, an extension of OT-CFM that aims to minimize the generation sensitivity to continuous conditions and demonstrate its competitiveness. Although MMOT-EFM is computationally expensive, the application of EFM will expand in the future as more efficient algorithms for MMOT are developed.

Figure 5: Results in § 7.2

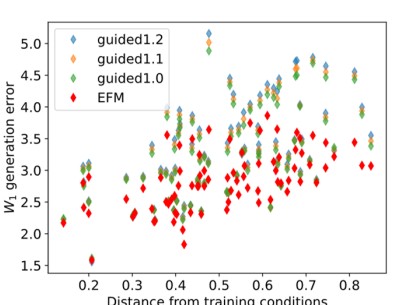

Table 2: MMOT-EFM vs. baselines in conditional molecular generations in § 7.3.

|  | Conditional Generation MAE |
| --- | --- |
| FM (Tong et al., 2023b) | $1.120 \pm 0.142$ |
| COT-FM (Chemseddine et al., 2024) | $0.966 \pm 0.122$ |
| **MMOT-EFM (ours)** | $\mathbf{0.918 \pm 0.122}$ |

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

## A    Mathematical description of Extended Flow Matching Theory

We aim to sample from the unknown conditional distribution $\Omega \ni c \mapsto p(\bullet \mid c) \in \mathcal{P}(D)$. We extend the flow matching technique developed in (Lipman et al., 2023) for this aim. The technique evolves unconditional probability distributions $\mu_t \in \mathcal{P}(D)$, $t \in [0, 1]$ from a source distribution $\mu_0$ (such as Gaussian $\mathcal{N}(,)$) to a target distribution $\mu_1 \approx p^{\mathrm{data}}$ by means of a continuity equation. We then introduce a generalized continuity equation that evolves conditional distributions $\mu_{t,c}$, $t \in [0, 1]$, $c \in \Omega$ from source distributions $\mu_0$ to the target distributions $\mu_{t=1,c} \approx p^{\mathrm{data}}(\bullet \mid c)$.

To realize this evolution, this section gives an example of how to construct a (at least approximate) solution of the generalized continuity equation and a design of the source distributions $\mu_{t=0,c}, c \in \Omega$.

### A.1    Notations

- $\langle \bullet, \bullet \rangle$ is the standard inner product and $|\bullet| := \sqrt{\langle \bullet, \bullet \rangle}$.

- $D \ni x = (x^1, \ldots, x^q)$; data space

- $t \in [0, 1]$; generation time

- $c \in \Omega \subset \mathbb{R}^p$; conditions in a bounded domain $\Omega$.

- $\xi = (\xi^0, \xi^1, \ldots, \xi^p) := (t, c) \in \widetilde{\Omega} := [0, 1] \times \Omega$.

- $x \in D \subset \mathbb{R}^q$; data in a compact subset $D$

- For a matrix-valued function $u \colon \Xi \times D \to \mathbb{R}^{d \times \dim \Xi}$, let $u_{i,j}$ denote its $(i, j)$-th coordinate, where $i \in [d]$, $j \in [\dim \Xi]$. We then define

$$\mathrm{div}_x\, u \colon \Xi \times D \to \mathbb{R}^{\dim \Xi} \quad \text{as} \quad \mathrm{div}_x\, u(\xi, x) := \left( \sum_{i=1}^{d} \partial_i u_{i,0}(\xi, x), \ldots, \sum_{i=1}^{d} \partial_i u_{i,\dim \Xi}(\xi, x) \right)^{\top} .$$
(A.1)

- For $\varphi \in C^1(\widetilde{\Omega} \times D; \mathbb{R}^{p+1})$,

$$\nabla_x \varphi := \begin{pmatrix} \partial_{x^1} \varphi^0 & \ldots & \partial_{x^1} \varphi^p \\ \vdots & \ddots & \vdots \\ \partial_{x^q} \varphi^0 & \ldots & \partial_{x^q} \varphi^p \end{pmatrix} \in \mathbb{R}^{q \times (p+1)}.$$

- $\mathcal{P}(X)$; the space of Borel probability measures on a space $X$, endowed with the narrow topology

- $\mathcal{P}_2(X)$; the $L^2$-Wasserstein space

- $\delta_x \in \mathcal{P}_2(X)$; the delta measure supported at $x \in X$

- $\mu_\bullet \colon \widetilde{\Omega} \ni \xi \mapsto \mu_\xi \in \mathcal{P}(D)$ conditional probability distribution

- $L^2(\Omega; X)$; the Lebesgue space valued in a metric space $X$, see (Lavenant, 2019, Definition 3.1)

- $H^1(\Omega; X)$; the Sobolev space valued in a metric space $X$, see (Lavenant, 2019, Definition 3.18). In particular, we set $\Gamma := H^1(\widetilde{\Omega}; D)$

- $\mathrm{Dir}(\mu)$ is the Dirichlet energy of $\mu \in L^2(\Omega; \mathcal{P}(D))$, see (Lavenant, 2019, Definition 3.5).

- $\mathrm{Unif}(S)$ is the uniform distribution on a subset $S$ of a Euclidean space with unit mass.

- $Q \in \mathcal{P}(\Psi)$. We will denote by $\psi$ the sample from a probability distribution $Q$.

- $\sigma(X)$ denotes the $\sigma$-algebra of a random variable

Following the notation in (Durrett, 2019), we also use the notation $x \sim p$ to designate that $x$ is sampled from the distribution $p$.

## A.2 GENERALIZED CONTINUITY EQUATION

According to (Lavenant, 2019, Definition 3.4), we introduce a distributional solution of a generalized continuity equation formally given as

$$\nabla_\xi \mu(\xi, x) + \mathrm{div}_x(\mu(\xi, x)v(\xi, x)) = 0. \tag{A.2}$$

The rigorous sense of (A.2) is stated in the following.

**Definition A.1** (A distributional solution of the generalized continuity equation). A pair $(\mu, v)$ of a Borel mapping $\mu \colon \widetilde{\Omega} \to \mathcal{P}(D)$ valued in probability measures and a Borel matrix field $v \colon \widetilde{\Omega} \times D \to \mathbb{R}^{q \times (p+1)}$ is a *solution of the continuity equation* if it holds that

$$\int_{\widetilde{\Omega}} \int_{\mathbb{R}^q} |v(\xi, x)|^2 \, \mathrm{d}\mu_\xi(x) \, \mathrm{d}\xi < +\infty,$$

and

$$\int_{\widetilde{\Omega}} \int_{\mathbb{R}^q} (\mathrm{div}_\xi \, \varphi(\xi, x) + \langle \nabla_x \varphi(\xi, x), v(\xi, x) \rangle) \, \mathrm{d}\mu_\xi(x) \, \mathrm{d}\xi = 0,$$

for all $\varphi \in C_c^\infty(\widetilde{\Omega} \times \mathbb{R}^q; \mathbb{R}^{p+1})$.

If a solution $(\mu, v)$ of the continuity equation is smooth, a path $\gamma$ on $\widetilde{\Omega}$ induces a path on $\mathcal{P}(D)$:

**Proposition A.2** (Lifting conditional paths to probability paths). *Let $(\mu, v)$ be a solution of the continuity equation and $\gamma \colon [0, 1] \ni s \mapsto \gamma(s) \in \widetilde{\Omega}$ be a continuously differentiable curve in $\widetilde{\Omega}$. Set $\mu^\gamma \coloneqq \mu_{\gamma(\bullet)} \colon [0, 1] \to \mathcal{P}(D)$ and $v^\gamma(s, x) \coloneqq v(\gamma(s), x)\dot\gamma(s) \in \mathbb{R}^q$ for $(s, x) \in [0, 1] \times \mathbb{R}^q$.*

*Suppose that $\mathrm{Dir}(\mu) < +\infty$ and there exists a probability density $\rho \in C^\infty(\widetilde{\Omega}; L^\infty(D))$ of $\mu$ with respect to the Lebesgue measure.*

*Then, $(\mu^\gamma, v^\gamma)$ satisfies the continuity equation in the sense of distributions, i.e.,*

$$\int_0^1 \int_{\mathbb{R}^q} (\partial_s \zeta(s, x) + \langle \nabla_x \zeta(s, x), v^\gamma(s, x) \rangle) \, \mathrm{d}\mu_s^\gamma(x) \, \mathrm{d}s = 0,$$

*for all $\zeta \in C_c^\infty([0, 1] \times \mathbb{R}^q)$.*

*Proof.* By (Lavenant, 2019, Proposition 3.16), there exists a unique $\varphi(\xi, \bullet) \in H^1(D; \mathbb{R}^{p+1})$ for every $\xi \in \overset{\circ}{\widetilde{\Omega}}$ satisfying

$$\nabla_\xi \rho(\xi, x) + \mathrm{div}_x(\rho(\xi, x)\nabla_x \varphi(\xi, x)) = 0, \ x \in \overset{\circ}{D},$$

and $v = \nabla_x \varphi$ on $\mathrm{supp}\,\mu$, where $\overset{\circ}{X}$ is the interior of a subset $X$. Thus, we have

$$\begin{aligned}
\partial_s \rho(\gamma(s)) + \mathrm{div}_x(\rho(\gamma(s), x)v^\gamma(s, x)) &= (\nabla_\xi \rho(\gamma(s), x) + \mathrm{div}_x(\rho(\gamma(s), x)v(\gamma(s), x)))\dot\gamma(s) \\
&= (\nabla_\xi \rho(\gamma(s), x) + \mathrm{div}_x(\rho(\gamma(s), x)\nabla_x \varphi(\gamma(s), x)))\dot\gamma(s) \\
&= 0.
\end{aligned}$$

$\blacksquare$

*Remark* A.3. The smoothness assumption of Proposition A.2 recommends us to use some smooth probability measures as source distributions $\mu_{t=1,c}$, $c \in \Omega$.

According to Proposition A.2 and the well-known fact (see (Ambrosio et al., 2008, Proposition 8.1.8)), if we want a sample under a certain condition $c \in \Omega$, we can flow samples from a source distribution according to the family $(v^\gamma(s, \bullet))_{s \in [0, 1]}$ of vector fields determined from a path $\gamma$ satisfying $\gamma(1) = (1, c)$.

## A.3 PRINCIPLED MASS ALIGNMENT

A straightforward generalization of (Kerrigan et al., 2024a, Theorem 1 and Theorem 3) yields the following principle in flow marching theory.

**Lemma A.4** (Principled mass alignment lemma). *Let $\mathcal{F}$ be a separable (complete) metric space and $P$ be a Borel probability measure on $\mathcal{F}$. Let $(\mu^f, v^f)$ be a solution of the continuity equation, in the sense of Definition A.1, for each $f \in \mathcal{F}$. Set the marginal distribution as*

$$\bar{\mu} := \int_{\mathcal{F}} \mu^f \, \mathrm{d}P(f).$$

*Assume that*

$$\iint_{\mathcal{F}\,\widetilde{\Omega}} \int_{\mathbb{R}^q} \left| v^f(\xi, x) \right|^2 \mathrm{d}\mu_\xi^f(x) \, \mathrm{d}\xi \, \mathrm{d}P(f) < +\infty,$$

*and $\mu_\xi^f$ is absolutely continuous with respect to $\bar{\mu}_\xi$ for $P$-a.e. $f$ and a.e. $\xi \in \widetilde{\Omega}$. Then, $(\bar{\mu}, \bar{v})$ is also a solution, where*

$$\bar{v}(\xi, x) = \int_{\mathcal{F}} v^f(\xi, x) \frac{\mathrm{d}\mu_\xi^f}{\mathrm{d}\mu_\xi}(x) \, \mathrm{d}P(f),$$

*for $(\xi, x) \in \widetilde{\Omega} \times D$. Moreover, for another matrix field $u$ satisfying*

$$\int_{\widetilde{\Omega}} \int_{\mathbb{R}^q} |u(\xi, x)|^2 \, \mathrm{d}\bar{\mu}_\xi(x) \, \mathrm{d}\xi < +\infty,$$

*we have*

$$\int_{\widetilde{\Omega}} \int_{\mathbb{R}^q} \langle \bar{v}(\xi, x), u(\xi, x) \rangle \, \mathrm{d}\bar{\mu}_\xi(x) \, \mathrm{d}\xi = \iint_{\mathcal{F}\,\widetilde{\Omega}} \int_{\mathbb{R}^q} \langle v^f(\xi, x), u(\xi, x) \rangle \, \mathrm{d}\mu_\xi^f(x) \, \mathrm{d}\xi \, \mathrm{d}P(f). \tag{A.3}$$

Lemma A.4 leads to Theorem 3.4 as follows: first, in Lemma A.4, identify $(\bar{v}, u)$ with $(u, u_\theta)$ in Theorem 3.4. hen we see from (A.3) that

- $\int_{\Xi} \mathbb{E}_{x \sim \mu_\xi} \left[ \langle u(\xi, x), u_\theta(\xi, x) \rangle \right] \mathrm{d}\xi$ and
- $\int_{\Xi} \mathbb{E}_{\psi \sim Q, x \sim \mu_\xi^\psi} \left[ \langle v^\psi(\xi, x), u_\theta(\xi, x) \rangle \right] \mathrm{d}\xi$ are equal,

where $v^\psi$ is a matrix field such that $v^\psi(\xi, \psi(\xi)) = \nabla_\xi \psi(\xi)$ with $\xi \in \Xi$. Also, because $\mu_\xi^\psi = \delta_{\psi(\xi)}$ is a delta distribution concentrated on $\psi(\xi)$, these are both equal to $\int_{\Xi} \mathbb{E}_{\psi \sim Q} \left[ \langle \nabla_\xi \psi(\xi), u_\theta(\psi(\xi)) \rangle \right] \mathrm{d}\xi$, as well. If we use this identity to the expansion of the square norm in (3.6), then the Theorem 3.4 follows from the same logic as (Kerrigan et al., 2024a, Theorem 3).

## A.4 LIFTING DATA-VALUED FUNCTION TO PROBABILITY-MEASURE-VALUED FUNCTION

In order to construct a solution of the generalized continuity equation, we start to consider a particle-based solution of the continuity equation.

According to (Brenier, 2003, Subsection 3.1) and (Lavenant, 2019, Section 5), we can easily construct a solution of the continuity equation from a given function $\psi \in H^1(\widetilde{\Omega}; D)$.

**Lemma A.5.** *Let $\psi \in H^1(\widetilde{\Omega}; D)$ be a function satisfying*

$$\int_{\widetilde{\Omega}} |\nabla_\xi \psi(\xi)|^2 \, \mathrm{d}\xi < +\infty.$$

*Set $\mu_\bullet^\psi := \delta_{\psi(\bullet)} \in H^1(\widetilde{\Omega}; \mathcal{P}(D))$. Assume that there exists a matrix field satisfying*

$$v^\psi(\xi, \psi(\xi)) = \nabla_\xi \psi(\xi), \tag{A.4}$$

*for $\xi \in \widetilde{\Omega}$. Then, $(\mu^\psi, v^\psi)$ is a solution of the continuity equation.*

Combining Lemmas A.4 and A.5, we can construct another solution of the continuity equation.

**Corollary A.6** (The paths make the solution.). *Let $Q \in \mathcal{P}(H^1(\widetilde{\Omega}; D))$ be a Borel probability measure, and $(\mu^\psi, v^\psi)$ be a solution defined in Lemma A.5 $Q$-a.e. $\psi \in H^1(\widetilde{\Omega}; D)$ and*

$$\mu^Q := \int\limits_{H^1(\widetilde{\Omega};D)} \mu^\psi \, \mathrm{d}Q(\psi)$$

*is their marginal distribution. Assume that*

$$\int\limits_{H^1(\widetilde{\Omega};D)} \int\limits_{\widetilde{\Omega}} \int\limits_{\mathbb{R}^q} \left| v^\psi(\xi, x) \right|^2 \mathrm{d}\mu^\psi_\xi(x) \, \mathrm{d}\xi \, \mathrm{d}Q(\psi) < +\infty,$$

*and $\mu^\psi \ll \mu^Q$. Then, $(\mu^Q, v^Q)$ is also a solution of the continuity equation, where*

$$v^Q = \int\limits_{H^1(\widetilde{\Omega};D)} v^\psi(\xi, x) \frac{\mathrm{d}\mu^\psi_\xi}{\mathrm{d}\mu_\xi}(x) \, \mathrm{d}Q(\psi) \,.$$

# B    TECHNICAL PROOFS

The following claim follows immediately from the convexity of the Dirichlet energy as shown in Lavenant (2019, Proposition 3.13) and from Jensen's inequality:

**Proposition B.1** (Straightness is controlled by $\psi$). *Let $\mu_{t,c} = \mathbb{E}_{\psi \sim Q} \left[ \delta_{\psi(t,c)} \right]$ $((t, c) \in I \times \Omega)$ with $\eta \in \mathcal{P}(D)$. Then, the Dirichlet energy of $\mu: I \times \Omega \to \mathcal{P}(D)$ is bounded as*

$$\mathrm{Dir}_{I \times \Omega}(\mu) \leq \iint\limits_{I \times \Omega} \mathbb{E}_{\psi \sim Q} \left\| \nabla_{t,c} \psi(t, c) \right\|^2 \mathrm{d}t \mathrm{d}c \,.$$

**Proposition B.2.** *Let $\mu \in H^1(\widetilde{\Omega}; \mathcal{P}(D))$ be a smooth solution of the continuity equation, and $v: \widetilde{\Omega} \times \mathbb{R}^q \to \mathbb{R}^{q \times (p+1)}$ is the matrix field associated with $\mu$. Assume that $v \in C^1(\widetilde{\Omega} \times \mathbb{R}^q; \mathbb{R}^{q \times (p+1)})$ and the derivatives $\partial_c v$, $\partial_x v$ of $v$ is bounded on $\widetilde{\Omega} \times \mathbb{R}^q$. Then, there exists a constant $C > 0$ depend on $p, q$ such that*

$$\mathrm{Dir}(\mu(1, \bullet)) \leq C \exp\left( \left\| \partial_x v \right\|_{L^\infty(\widetilde{\Omega} \times \mathbb{R}^q; \mathcal{B}(\mathbb{R}^q \times \widetilde{\Omega}; \mathbb{R}^q))} \right) (\mathrm{Dir}(\mu(0, \bullet)) + \left\| \partial_c v \right\|_\infty).$$

*Here, $\|f\|_\infty = \sup_{(\xi, x) \in \widetilde{\Omega} \times \mathbb{R}^q} |f(\xi, x)|$ for a finite-dimensional valued continuous function $f$ on $\widetilde{\Omega} \times \mathbb{R}^q$.*

The proof of Proposition B.2 is similar to (Isobe, 2023, Proposition 5.4).

***Proof.*** By virtue of (Lavenant, 2019, Proposition 3.21), we have to estimate

$$\mathrm{Dir}(\mu(1, \bullet)) = \lim_{\varepsilon \to 0} \frac{C_p}{\varepsilon^{p+2}} \iint_{\Omega^2} W_2^2(\mu(1, c^1), \mu(1, c^2)) \, \mathrm{d}c^1 \mathrm{d}c^2 \,.$$

The integrand of the above is decomposed as

$$
\begin{aligned}
W_2(\mu(1, c^1), \mu(1, c^2)) &= W_2\left( \Phi^{1,c^1}_\# \mu(0, c^1), \Phi^{1,c^2}_\# \mu(0, c^2) \right) \\
&\leq W_2\left( \Phi^{1,c^1}_\# \mu(0, c^1), \Phi^{1,c^2}_\# \mu(0, c^1) \right) + W_2\left( \Phi^{1,c^2}_\# \mu(0, c^1), \Phi^{1,c^2}_\# \mu(0, c^2) \right).
\end{aligned}
$$
(B.1)

Here $\Phi^{t,c}: \mathbb{R}^q \to \mathbb{R}^q$ is a flow mapping satisfying

$$\Phi^{t,c}(x) = x + \int_0^t v(s, c, \Phi^{t,c}(x)) \begin{pmatrix} 1 \\ 0 \end{pmatrix} \mathrm{d}s \,.$$

The first term of (B.1) is bounded as

$$W_2\left(\Phi_\#^{1,c^1}\mu(0,c^1), \Phi_\#^{1,c^2}\mu(0,c^1)\right)^2 \le \int_{\mathbb{R}^q}\left|\Phi^{t,c^1}(x) - \Phi^{t,c^2}(x)\right|^2 \mathrm{d}\mu_{0,c^1}(x).$$

Then, the integrand is also bounded by

$$\left|\Phi^{t,c^1}(x) - \Phi^{t,c^2}(x)\right| \le \int_0^t \left\|v(s,c^1,\Phi^{s,c^1}(x)) - v(s,c^2,\Phi^{s,c^2}(x))\right\|_{\mathrm{op}} \mathrm{d}s$$

$$\le |c^1 - c^2|\|\partial_c v\|_\infty$$

$$+ \int_0^t \|\partial_x v\|_\infty \left|\Phi^{t,c^1}(x)) - \Phi^{t,c^2}(x))\right| \mathrm{d}s.$$

Thus, the Gronwall inequality yields

$$\left|\Phi^{t,c^1}(x) - \Phi^{t,c^2}(x)\right| \le |c^1 - c^2|\|\partial_c v\|_{L^\infty(\widetilde{\Omega}\times\mathbb{R}^q;\mathcal{B}(\Omega\times\widetilde{\Omega};\mathbb{R}^q))} \exp\left(\|\partial_x v\|_{L^\infty(\widetilde{\Omega}\times\mathbb{R}^q;\mathcal{B}(\mathbb{R}^q\times\widetilde{\Omega};\mathbb{R}^q))}\right).$$
(B.2)

By a similar argument, the second term of (B.1) is also bounded as

$$W_2\left(\Phi_\#^{1,c^2}\mu(0,c^1), \Phi_\#^{1,c^2}\mu(0,c^2)\right) \le W_2(\mu(0,c^1),\mu(0,c^2))\exp\left(\|\partial_x v\|_{L^\infty(\widetilde{\Omega}\times\mathbb{R}^q;\mathcal{B}(\mathbb{R}^q\times\widetilde{\Omega};\mathbb{R}^q))}\right).$$
(B.3)

Combining (B.2) and (B.3) completes the proof. ∎

## C PSEUDO-CODES

---
**Algorithm 4** Algorithm of OT-CFM

---
**Input:** Neural Network $v_\theta: I \times D \to \mathbb{R}^d$, the source distribution $\mu_0$, the dataset $D_* \subset D$ from a target distribution $\mu$.
**Return:** $\theta \in \mathbb{R}^p$
 1: **for** each iteration **do**
       # Step 1:  Sample from datasets
 2:    Sample a batch $B^0$ from $\mu_0$
 3:    Sample a batch $B^1$ from $D_*$
       # Step 2:  Construct $\psi: I \to D$
 4:    Construct an optimal transport plan $\pi$ between $B^0$ and $B^1$
 5:    Jointly sample $(x_0, x_1) \sim \pi$
 6:    Sample $t \sim \mathrm{Unif}(I)$
 7:    Compute
$$\psi_t := \psi(t \mid x_0, x_1)$$
$$= (1-t)x_0 + tx_1$$
$$\dot\psi_t := \dot\psi(t \mid x_0, x_1)$$
$$= x_1 - x_0$$
 8:    Update $\theta$ by the gradient of $\|v_\theta(t,\psi_t) - \dot\psi_t\|^2$
 9: **end for**

---

## D SAMPLING OF $\bar\psi$ IN (4.1) IN § 4 FOR MMOT-EFM

In this section, we follow the notation in § 4 and describe in more detail the construction of $\bar\psi(c|\boldsymbol{x}_{C_0})$ in (4.1), which is

$$\psi(t,c \mid x_{0,c}, \boldsymbol{x}_{C_0}) = (1-t)x_{0,c} + t\bar\psi(c \mid \boldsymbol{x}_{C_0})$$

---

**Algorithm 5** Flow Matching (Training)

---

**Input:** Neural Network $v_\theta \colon I \times D \to \mathbb{R}^d$, the source distribution $\mu_0$, the dataset $D_* \subset D$ from a target distribution $\mu$.

**Return:** $\theta \in \mathbb{R}^p$

1: **for** each iteration **do**

   `# Step 1:  Sampling from datasets`

2:   Sample batches $B^0 = \{x_0^i\}_{i=1}^N$ from source $p_0$

3:   Sample batches $B^1 = \{x_1^j\}_{j=1}^N$ from dataset $D_*$

   `# Step 2:  Constructing a supervisory path` $\psi$

4:   Construct an optimal transport plan $\pi \in \mathbb{R}^{N \times N}$ between $B^0$ and $B^1$

5:   Jointly sample $(x_0, x_1) \in B^0 \times B^1$ from $\pi$

6:   Sample $t \in I$

7:   Compute
   (A) $\psi_t := \psi(t \mid x_0, x_1) = (1-t)x_0 + tx_1$
   (B) $\nabla \psi_t := \nabla_t \psi(t \mid x_0, x_1) = x_1 - x_0$

   `# Step 3:  Learning vector fields`

8:   Update $\theta$ by the gradient of $\|v_\theta(t, \psi_t) - \nabla \psi_t\|^2$

9: **end for**

---

**Algorithm 6** `ODEsolve` for generation

---

**Input:** Initial data $x_0 \in D$, vector fields $v \colon I \times D \to \mathbb{R}^d$

**Return:** Terminal value $\phi_1^v(x_0)$ of the solution of ODE $\dot{\phi}_t^v(x_0) = v(t, \phi_t^v(x_0))$

1: Compute $\phi_1(x_0)$ via a discretization of the ODE in $t$

---

**Algorithm 7** Extended Flow Matching (Training)

---

**Input:** Condition set $C \subset \Omega \subset \mathbb{R}^k$, set of datasets $D_c \subset D \subset \mathbb{R}^d$ for each $c \in C$, network $u_\theta \colon I \times \Omega \times D \to \mathbb{R}^{d \times (1+k)}$, source distributions $p_0(\cdot \mid c)$ $(c \in C)$

**Return:** $\theta \in \mathbb{R}^p$

1: **for** each iteration **do**

   `# Step 1:  Sampling from datasets`

2:   Sample $C_0 = \{c_i\}_{i=1}^{N_c} \subset C$

3:   Sample a batch $B_{0,c}$ from $p_0(x \mid c)$ for each $c \in C_0$

4:   Sample a batch $B_{1,c}$ from $D_c$ for each $c \in C_0$

5:   Put $B^0 := \{B_{0,c}\}_{c \in C_0}$ and $B^1 := \{B_c\}_{c \in C_0}$

   `# Step 2:  Constructing supervisory paths` $\{\psi_j\}_{j=1}^N$

6:   Construct a transport plan $\pi$ among $B^0$ and $B^1$

   `# see § 4`

7:   Sample $\{(x_{t,c}^j)_{(t,c) \in \{0,1\} \times C_0}\}_{j=1}^N \subset D^{2N_c}$ from $\pi$

8:   For all $j \in [1:N]$, define $\psi_j \colon I \times \Omega \to D$ that regresses $(x_{t,c}^j)_{(t,c) \in \{0,1\} \times C_0}$ on $\{0,1\} \times C_0$
   `# see Equation (4.1)`

9:   Sample $\{t_k\}_{k=1}^{N_t} \subset I$

10:   Sample $\{c_l'\}_{l=1}^{N_c'} \subset \mathrm{Conv}(C_0)$

11:   For all $j \in [1:N]$, $k \in [1:N_t]$, $l \in [1:N_c']$, compute
   (A) $\psi_{j,k,l} := \psi_j(t_k, c_l')$
   (B) $\nabla \psi_{j,k,l} := \nabla_{t,c} \psi_j(t_k, c_l')$

   `# Step 3:  Learning matrix fields`

12:   Compute the loss

$$L(\theta) = \frac{1}{N N_t N_c'} \sum_{j,k,l} \|u_\theta(t_k, c_l', \psi_{j,k,l}) - \nabla \psi_{j,k,l}\|^2$$

13:   Update $\theta$ by the gradient of $L(\theta)$

14: **end for**

---

and the corresponding joint distribution of $\boldsymbol{x}_{C_0} := \{x_i\}_{c_i \in C_0}$ on $D^{2|C_0|}$ we used in step 2 of the training algorithm. In the final part of this section, we also elaborate how we couple $x_{0,c}$ with $\boldsymbol{x}_{C_0}$.

As we describe in the main manuscript, we introduce our EFM as a direct extension of FM as a method to transform one distribution to another through a learned vector field. In particular, we present in this paper an implementation of EFM which extends OT-CFM Tong et al. (2023b), which aims to train FM as an approximate optimal transport between two distributions (source $\mu_0$ and target $\mu_1$). To formalize this extension, we need to desribe OT as a minimization of Dirichlet Energy.

### D.1 OT-CFM AS APPROXIMATE DICIRHLET ENERGY MINIMIZATION

As is principally described in Lavenant (2019), OT emerges as a coupling of the source $\mu_0$ and the target $\mu_1$ constructed from the constant-speed geodesic (with respect to Wasserstein distance) between $\mu_0$ and $\mu_1$, which can be realized by minimizing the Dirichlet energy

$$\text{Dir}(\mu) = \inf_{v:I \times D \to \mathbb{R}^d} \left\{ \int_{[0,1] \times D} \frac{1}{2} \|v(t,x)\|^2 \mu_t(\mathrm{d}x)\mathrm{d}t \,\middle|\, \partial_t \mu_t(x) + \text{div}_x(\mu_t(x)v(t,x)) = 0 \right\} \quad \text{(D.1)}$$

over all set of $\mu:[0,1] \to \mathcal{P}(D)$ satisfying $\mu(0) = \mu_0$, $\mu(1) = \mu_1$. It is well known that in the standard Euclidean metric space, the minimal energy is achieved by $\mu$ corresponding to $v(t,x)$ that is the derivative of a straight-line of form $\psi^T(t \mid x) = tT(x) + (1-t)x$ where $T: D \to D$, and more particularly as the minimum of

$$\int_{D \times D} \frac{1}{2} \|x - y\|^2 \pi(\mathrm{d}x, \mathrm{d}y) = \int_D \frac{1}{2} \|\partial_t \psi^T(t|x)\|^2 (I \times T)_\# \mu_0(dx) \quad \text{(D.2)}$$

over all $\pi \in \mathcal{P}(D \times D)$ with marginal distribution $\mu_0$ and $\mu_1$ or equivalently over all $T$ with $T\#\mu_0 = \mu_1$. In OT-CFM, this $\pi$(or $T$) is approximated by the discrete optimal transport solution over a pair of batches $B_0, B_1$ sampled respectively from source and target distributions. Note that, in this view, $(I \times T)_\# \mu_0$ induces a distribution $Q$ on the path $[0,1] \to D$ generating $\psi^T(t|x)$ with randomness derived from $x$.

Theorem 3.1 of Yim et al. (2024) guarantees that the (batch)sample-averaged version of $\mu$ and the (batch)sample-averaged version of $v$ satisfies the continuity equation, thereby yielding the approximation of the dirichlet energy minimizing flow map.

### D.2 MMOT-EFM AS APPROXIMATE DICIRHLET ENERGY MINIMIZATION

To mimic this construction in multi-marginal setting of EFM, we aim to approximate the solution to the minimization of

$$\text{Dir}(\mu) = \inf_{v:\Omega \times D \to \mathbb{R}^{d \times k}} \left\{ \int_{\Omega \times D} \frac{1}{2} \|v(c,x)\|^2 \mu_\xi(\mathrm{d}x)\mathrm{d}c \,\middle|\, \partial_c \mu_\xi(x) + \text{div}_x(\mu(c,x)v(c,x)) = 0 \right\}$$
$$\text{(D.3)}$$

over all set of $\mu:\Omega \to \mathcal{P}(D)$ satisfying $\mu(c_i) = \mu_i$ for all $c_i \in C_0$. Note that when $\Omega = [0,1]$, this minimization problem (i.e. Dirichlet Problem) agrees with that of the OT problem on which the method of FM is established.

Now, in a similar philosophy as FM, we would aim to approximate this Dirichlet energy through multi-marginal optimal transport Piran et al. (2024) over discrete samples. Now, under *sufficient* regularity condition (Prop 5.6 Lavenant (2019)), we can similarly argue that there exists some probability $Q$ on the space $\mathcal{F} = H^1(\Omega, D)$ of a map from "condition" to "data" satisfying

$$\text{Dir}(\mu) = \int_{\Omega \times \mathcal{F}} \|\partial_c \psi(c)\|^2 Q(\mathrm{d}\psi)\mathrm{d}c \quad \text{(D.4)}$$

and our goal winds down to finding the energy-minimizing distribution $Q$. In this endeavor, we implicitly find $Q$ by specifying a particular space of functions $\mathcal{F}$ and generating $\psi: \Omega \to D$ from

a set of $\{(c_i, x_i)\}_{c_i \in C_0}$ of "condition value" and "observation" for jointly sampled $\{x_i\}_i$ as the regression

$$\bar{\psi}(\cdot|\{x_i\}_i) = \arg\min_{\psi \in \mathcal{F}} \sum_{c_i \in C_0} \|\psi(c_i) - x_i\|^2 \tag{D.5}$$

and minimize the energy with respect to the joint distribution $\pi$ on $D^{|C|}$ from which to sample $\{x_i\}_i$. That is, we aim to minimize

$$\int \|\nabla_c \bar{\psi}(c|\{x_i\}_i)\|^2 \pi(\{dx_i\}_i) \mathrm{d}c \tag{D.6}$$

with respect to $\pi$. This, indeed, is in the format of MMOT problem, where $c(\{x_i\}_i) \coloneqq \|\nabla_c \psi(c|\{x_i\}_i)\|^2$. $\mathcal{F}$ can be chosen, for example, as an RKHS or a space of linear function, so that the regression can be solved analytically with respect to $c$.

Just as is done in OT-CFM, we approximate this $\pi$ with the joint distribution over a finite tuple of batches $\{B_i\}_i$ with each $B_i$ sampled from $\mu_i$ corresponding to condition $c_i$. This approximation is indeed the very $\pi$ that we adopt in MMOT version of our EFM in step 2.

Now, by the virtue of Theorem of principle-mass-alignment A.6, we can argue that the (batch)sample-averaged distributions $\mu^\psi$ and the (batch)sample-averaged $v^\psi = \partial_c \psi$ solve the *generalized* continuity equation, thereby yielding the approximation of the Dirichlet energy minimizing map $\mu : \Omega \to \mathcal{P}(D)$.

Note that the above constructions of $\psi \sim Q$ is in complete parallel with that of OT-CFM. See Table 3 for the correspondences. We also note that this argument can be extended to $\tilde{\Omega} = [0, 1] \times \Omega$ in place

Table 3: OT-CFM vs MMOT-EFM

| Framework | OT-CFM | MMOT-EFM |
|---|---|---|
| $\mu$ | $[0, 1] \to \mathcal{P}(D)$ | $\Omega \to \mathcal{P}(D)$ |
| $\psi$ | $[0, 1] \to D$ | $\Omega \to D$ |
| $v$ | $\partial_t \psi$ | $\nabla_c \bar{\psi}$ |
| $(\mu, v)$ relation | Continuity | Generalized Continuity |
| Boundaries | $\{\mu_0, \mu_1\}$ | $\{\mu_i\}_{c_i \in C_0}$ |
| Approximation | OT | MMOT |

of $\Omega$. However, because of the computational cost of MMOT, we construct our generative model from (4.1), which combines $\bar{\psi}$ and the OT-CFM construction. In the next section, we elaborate on the construction of the approximation of $\pi$ in (D.6) from which to sample $\bar{\psi}$ in (4.1)

## D.3 Approximating MMOT

In general, MMOT is computationally heavy, and even with the advanced methods like the multi-marginal Sinkhorn method developed in (Lin et al., 2022), the computational cost scales as $|B|^{|C|}$, where $|B|$ is the batch size and $|C|$ is the number of conditions to be simultaneously considered. To reduce this cost, we took the approach of approximating MMOT through clustering. More particularly, when a batch from $B_i$ is sampled each from $\mu_i$ for condition $c_i$, we applied $K$-means nearest neighborhood clustering (KNN) to $B_i$, yielding sub-batches $\{U_{ik}\}_{c_i \in C_0, k \in [1:K]}$ with mean values $\{m_{ik}\}_{c_i \in C_0, k \in 1:K}$, where $\cup_{k \in 1:K} U_{ik} = B_i$. Let $M_i = \{m_{ik}\}_{k \in [1:K]}$ be the set of cluster-means for batch $i$. Instead of conducting MMOT directly on batch $B_i$, we conduct the MMOT on $\{M_i\}_i$, whose cost will be on the order of $K^{|C|}$. Applying $\arg\max$ operations on the result of MMOT from methods like the Sinkhorn method, we can obtain the deterministic coupling $\pi_m = (\bigtimes_i T_i)_{\#} \mathrm{Unif}(M_0)$ where $\mathrm{Unif}(M_0)$ is the uniform distribution on $M_0$. After sampling $m_{0k^*} \sim \mathrm{Unif}(M_0)$, we couple $U_{iT_i(k^*)}$ with a method of user's choice, where $T_i(k^*)$ is an *abuse of notation* satisfying

$$m_{iT_i(k^*)} = T_i(m_{0k^*}).$$

In our implementation of MMOT-EFM, we coupled $\{U_{iT_i(k^*)}\}_i$ with generalized-geodesic coupling as is used in Fan & Alvarez-Melis (2023), with center distribution being the standard Gaussian with

mean being the average of $\{U_{iT_i(k^*)}\}_i$. Although we provide a brief description of generalized-geodesic in § E, we would like to refer to Ambrosio et al. (2008) for a more thorough study.

Below, we summarize the sampling procedure of of $\{x_i\}_{c_i \in C_0}$ in $\psi(\cdot | \{x_i\}_{c_i \in C_0})$ of MMOT-EFM.

---

**Algorithm 8** MMOT sampling with Cluster

---

**Input:** Set of batches $\{B_i\}_i$ with each $B_i$ sampled from $p(\cdot | c_i)$
**Return:** Joint sample $\{x_i\}_i$ from $\{B_i\}_i$
     # Step 1: Cluster MMOT setup
1: Cluster each $B_i$ as $\cup_{k \in [1:K]} U_{ik} = B_i$ with $\mathrm{mean}(U_{ik}) = m_{ik}$
2: Set $M_i = \{m_{ik}\}_{k \in [1:K]}$
3: Use MMOT to produce coupling on $\{M_i\}_i$ via $\{T_i\}_i \# \mathrm{Unif}(M_0)$
     # Step 2: Sampling
4: Sample $m_{0k^*}$ from $\mathrm{Unif}(M_0)$
5: Compute $m_{iT_i(k^*)} := T_i(m_{0k^*})$
6: Jointly sample from $\{U_{iT_i(k^*)}\}$ with the method of user's choice, preferably with deterministic coupling, such as another round of MMOT or generalized-geodesic.

---

### D.4 COUPLING OF $\{x_{0,c_i}\}_{c_i \in C_0}$ AND $\{x_i\}_{c_i \in C_0}$

Ideally, it is more closely aligned with the theory of Dirichlet energy to include the source distributions $\{\mu(0, c_i)\}_i$ into the set of distributions to be coupled in the MMOT, and enact the argument in § D.2 with $\tilde{\Omega} = [0, 1] \times \Omega$ in place of $\Omega$. As mentioned in the previous section, however, the cost of empirical MMOT scales exponentially with the number of distributions to couples. We, therefore, took an alternative coupling strategy as a computational compromise.

First, recall from the step 1 of § 4 that $\{x_{0,c_i}\}_{c_i \in C_0}$ are already coupled with common standard Gaussian sample in the form of $\mu_{0,c} = \mathrm{Mean}[D_c] + \mathcal{N}(0, I)$. To couple $\{x_{0,c_i}\}_{c_i \in C_0}$ with $\{x_i\}_{c_i \in C_0}$ which are deterministically coupled through the routine of Section D.3 as $\{x_i\}_{c_i \in C_0} = \{\mathcal{T}_i(x_0)\}_{c_i \in C_0}$ with $x_0$ sampled from $p(\cdot | c_0)$, we may simply couple $x_{0,c_0}$ with $x_0$ and this will automatically induce the deterministic coupling of $\{x_{0,c_i}\}_{c_i \in C_0}$ and $\{x_i\}_{c_i \in C_0}$. In particular, if $B_{0,c_0}$ is a batch of samples from $p_0(\cdot | c_0)$ and $B_{1,c_0}$ is a batch of samples from $D_{c_0}$ in the step1 of the training, we may couple $B_{0,c_0}$ with $B_{1,c_0}$ with optimal transport with the methods of user's choice, such as those provided in Flamary et al. (2021).

## E A REMARK ON GENERALIZED GEODESIC COUPLING(GGC) AND THE SAMPLING OF $\bar{\psi}$ IN (4.1) IN § 4 FOR GGC-EFM

As we have mentioned in Section 3.1, EFM can be defined with any distribution $Q \in \mathcal{P}(\Psi)$ on the space of functions $\Psi := \{\psi \colon I \times \Omega \to D \mid \psi \text{ is differentiable}\}$ satisfying the boundary conditions (3.3). We also present still another construction of $\bar{\psi}$ derived from different coupling.

### E.1 GENERALIZED GEODESIC COUPLING

Generalized geodesic of $\{\mu_i\}$ with base $\nu \in \mathcal{P}(D)$, also known in the name of linear optimal transport Moosmüller & Cloninger (2020) in mathematical literatures, was introduced in (Ambrosio et al., 2008) as

$$\rho_a := \left( \sum_{i=1} a_i T_i \right)_{\#} \nu, \quad a \in \Delta_{m-1} \tag{E.1}$$

where $T_i$ is the optimal map from $\nu$ to $\mu_i$ and $\Delta_{m-1}$ is the set of all $\{a_i\}_{i=1}^m$ with $\sum_i a_i = 1$. This is indeed one of the generalizations to the McCann's interpolation used in OT between $\mu_0$ and $\mu_1$ through the expression

$$\rho_t := ((1 - t) \, \mathrm{Id} + tT)_{\#} \mu_0, \ t \in [0, 1]$$

which runs along the geodesic in $\mathcal{P}(D)$ with respect to Wasserstein distance. Note that $\rho_a$ in Generalized Geodesic provides not only provides deterministic coupling of $\{\mu_i\}$ through $\rho_{e_i} = T_{i\#}\nu = \mu_i$, it also interpolates unknown distributions for any $a \in \Delta_{m-1}$. We would refer to the deterministic coupling in the form of $T_{i\#}\nu = \mu_i$ as GGc-coupling.

### E.2 GGc sampling of $\bar{\psi}$

In analogy to the sampling procedure of $\bar{\psi}(\cdot \mid \{x_i\}_i)$ in MMOT-EFM with MMOT-coupled $\{x_i\}_i$, we may sample $\bar{\psi}(\cdot \mid \{x_i\}_i)$ with $\{x_i\}_i$ that is jointly sampled with GGc-coupling. We emphasize that $\bar{\psi}$ constructed in such a way does not necessarily minimize an explicit objective as Dirichlet energy and this might result in EFM with a somewhat erratic style transfer. For more empirical investigations, please see the main manuscript.

## F    Experiment details for conditional molecular generation

### F.1    Metrics

To evaluate our conditional generation, we use the pre-trained VAE model to encode EFM-generated latent vectors into molecular structures and compute the Mean Absolute Error(MAE) between the generated molecule's property values and the conditioning property values. MAEs are calculated separately for interpolation and extrapolation. All MAEs are first calculated for each property and then averaged for both properties.

### F.2    Dataset and baselines

We first trained a Site-information-encoded Junction Tree Variational Autoencoder (SJT-VAE) model, a variant implementation of the Junction Tree Variational Autoencoder (JT-VAE) (Jin et al., 2018). SJT-VAE was initially designed to eliminate the arbitrariness of JT-VAE and enable applications such as RJT-RL (Ishitani et al., 2022). We chose SJT-VAE over JT-VAE due to its superior reconstruction accuracy and faster training times. However, we expect that similar results could be reproduced with the original JT-VAE implementation.

Our SJT-VAE model was trained on the ZINC-250k dataset (Gómez-Bombarelli et al., 2018; Akhmetshin et al., 2021). A random subset of $80,000$ molecules was labeled with the number of HBAs and the number of rotatable bonds, with all labels computed using RDKit. These $80,000$ molecules were then binned into a 2D matrix based on their property values. From this matrix, we selected a region with concentrated data: molecules with $2$ and $4$ rotatable bonds and $3$ and $5$ HBAs, forming $4$ bins with property sets $(2,3)$, $(2,5)$, $(4,3)$, and $(4,5)$. To balance the dataset, we up-sampled or capped the number of training examples to $5,000$ per bin.

To evaluate out-of-distribution conditional generation, we generated molecules with property sets not included in the training set, specifically $(3,4)$, $(2,4)$, $(4,4)$, $(3,3)$, and $(5,5)$. For property sets where only one property is out-of-distribution, we calculated the MAE based solely on the out-of-distribution property.

All flow matching-based models, including MMOT-EFM and baselines, are trained with a batch size of 250 and the learning rate of $1\text{e}^{-4}$ for $160,000$ iterations. Training on a single Nvidia V-100 GPU with evaluation every 5000 iterations took around 4 hours.

## G    Computational Resources

All models were trained on a single Nvidia V100-16G GPU, and 100 epochs were completed within 4 hours. Training for the MMOT-EFM model is performed on a single Nvidia V100-16G GPU within 2.5 hours. The results of MMOT-EFM for synthetic experiments were yielded from a model trained over 100000 iterations in 5 hours.

## H    Additional figures

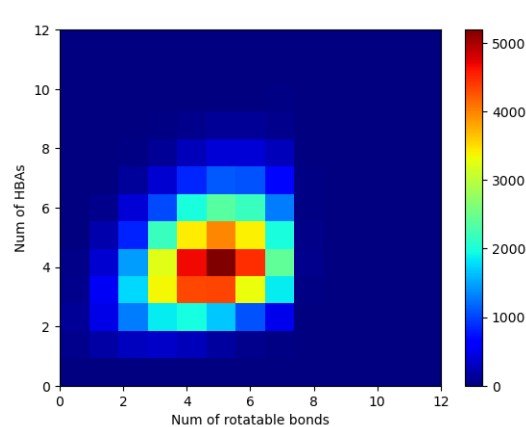

Figure 6: Training set rotatable bonds and HBAs label distribution

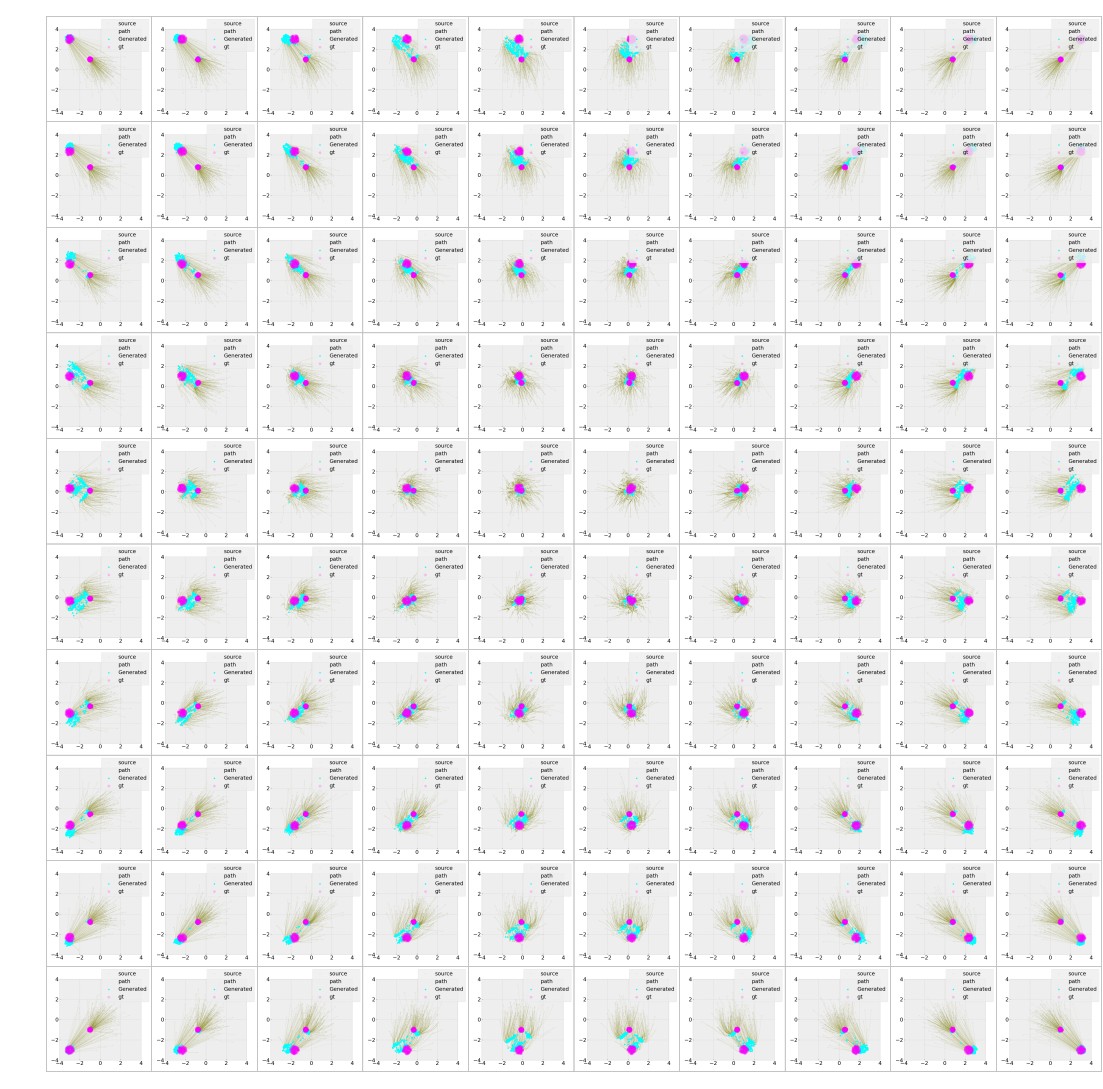

Figure 7: Conditional generation of the synthetic dataset by FM, organized in the grid for two axes of conditions.

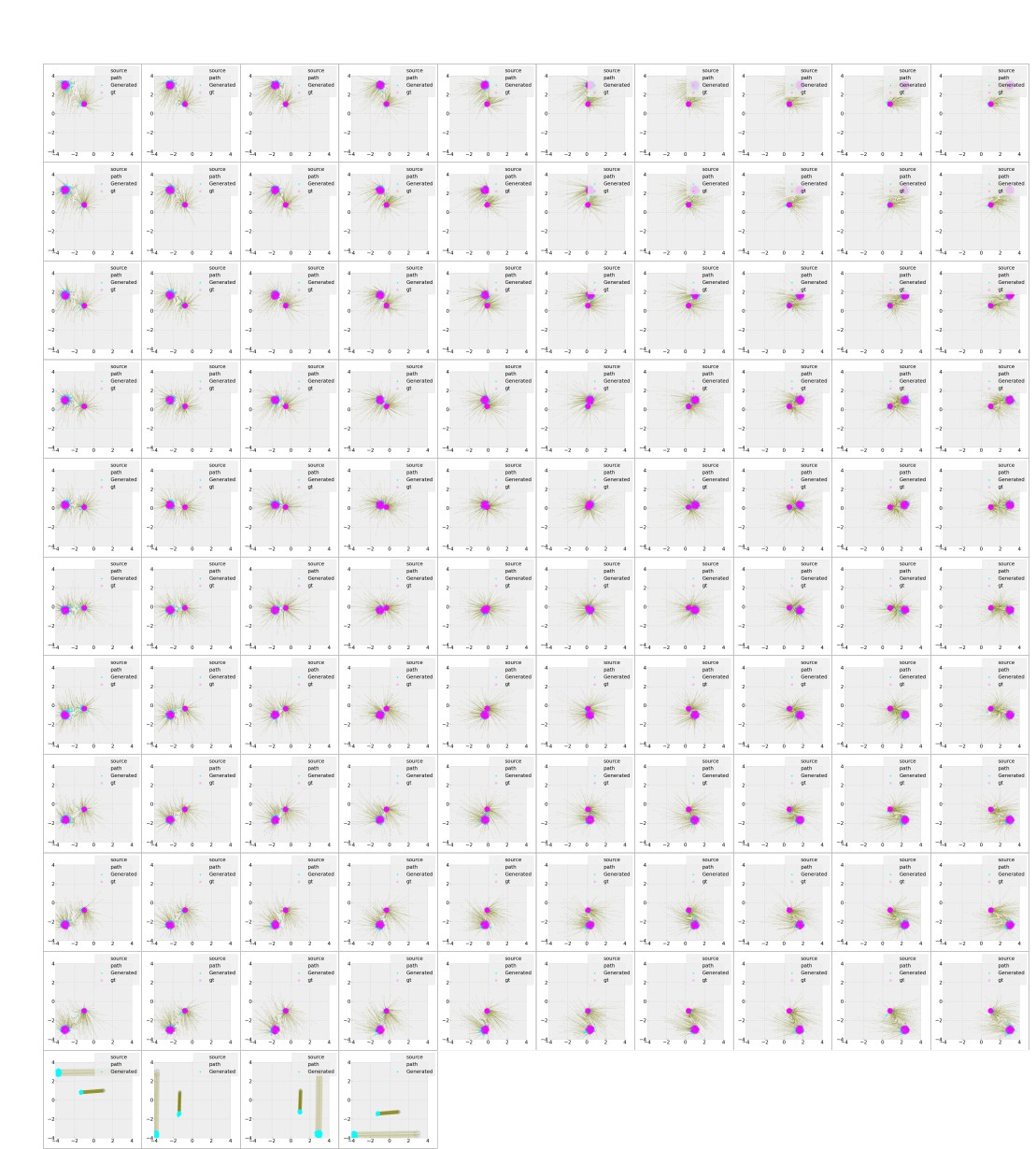

Figure 8: Conditional generation of the synthetic dataset by MMOT-EFM, organized in the grid for two axes of conditions. The figures in the bottom row are the result of style transfer.

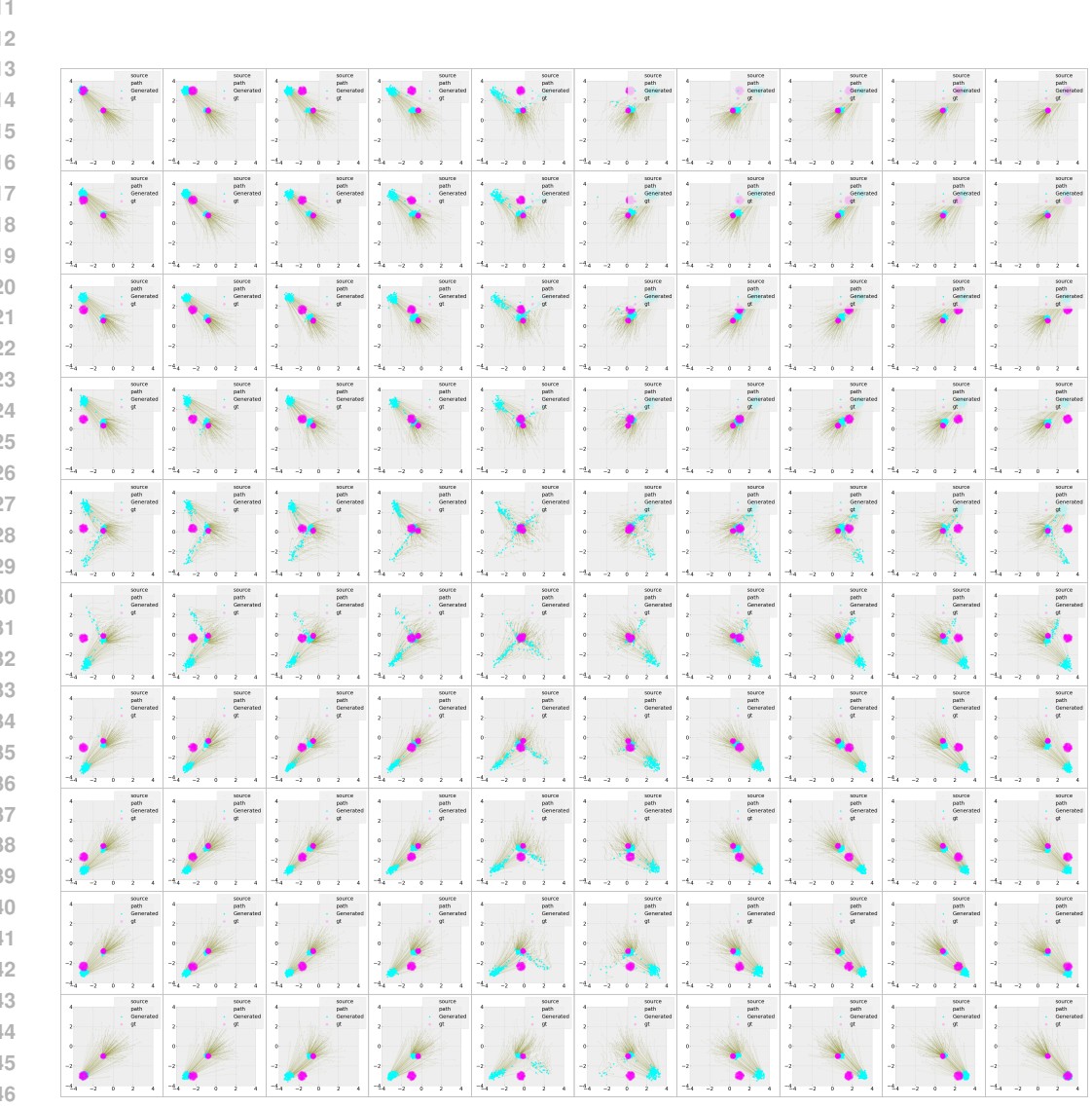

Figure 9: Conditional generation of synthetic dataset by Baysian(COT)-FM with $\beta = 10^2$, organized in grid for two axis of conditions.

