# OpenReview forum: "Extended Flow Matching  : a Method of Conditional Generation with Generalized Continuity Equation"
_ICLR.cc/2025/Conference — Submitted to ICLR 2025_

### Official Review · Reviewer_a59p · 2024-10-19

**Soundness:** 3
**Presentation:** 2
**Contribution:** 2
**Rating:** 5
**Confidence:** 5

**Summary:**

This paper introduces extended flow matching a new flow matching based method, that is designed for conditional generation. For this, the authors make use of the generalized continuity equation by Lavenant. The authors show that their proposed loss indeed has the correct gradients, i.e., regresses onto the true velocity field of the generalized continuity equation. The algorithm consists "learning" an interpolation via kernel regression (which is needed since "straight paths" are not the only viable solution anymore), and then regressing onto a flow matching loss where the is now matrix-valued. This is a generalization of the usual inverse problems framework of flow matching. Further, the authors showcase the effiacy of their algorithm via a toy example and conditional molecular generation.

**Strengths:**

I find the motivation very clear: Sometimes we already know the posteriors for several conditions (for instance in molecular dynamics, where we obtain some posterior samples via MCMC), and want to "smartly" interpolate between the conditions, i.e., learn a generative model which walks along "generalized "geodesics in the space of probability measures. I also like that the authors were very rigorous in their theorems and motivation for the developed algorithm.

**Weaknesses:**

However, the glaring weakness is that there is not clear cut numerical use case shown. I would like to see a not toyish example where we actually need several conditions and the transport between them. Usually, in the classical inverse problems works there is an implicit geodesic path taken where $y_t = t y + (1-t)y$, since one does not need to alter the condition if posterior sampling is the ultimate goal. If one wants to do style transfer (which seems to be the second motivation of this paper), then one can simply use a conditional FM network which receives the two conditions (source and target) as inputs. Therefore, while theoretically neat I am not convinced of why the generalized continuity equation and a network which moves efficiently also in the condition space, is advantageous. The authors can convince me by providing a clear example where either i) the classical conditional algorithms are not applicable or ii) this approach significantly outperforms the other flow matching models.

I also have some smaller concerns.

1) The scaling in $N_c$ and condition dimension seems to be bad. can you provide the run times for the molecular example also for the baselines? it only says in the appendix that they were completed within 4 hours, but I expect the baselines to train much quicker. Also latent space of a VAE is pretty low dimensional. Please provide training your conditional flow matching model on MNIST (no VAEs..), where the condition space is not discrete (i.e., for instance inpainting). Even if this does not fit your motivation, I would like to see the results in such a more standard example and this would improve my confidence in the scalability.

2) Appendix D5 and F are empty (or almost empty).

3) you do not seem to provide any code. I find the algorithm description to be not perfectly clear, there I would very strongly suggest that you at least publish code for the toy example.

4) I believe that the example 7.1 is meaningless. You construct a random example with sparse conditions. Then you show, that your algorithm performs better on the OOD. But basically you can construct an inverse problem which aligns with your in distribution posteriors and does anything else on the OOD data. Of course I am aware that your point is that your algorithm is minimizing the Dirichlet energy and you measure the distribution induced by this. However, it is not clear to me if this is the theoretically optimal thing to do (wrt to Wasserstein). I am guessing that your algorithm computes something like Wasserstein barycenters weighted by some distance to the known conditions? Please clarify why the minimization of the generalized Dirichlet energy should yield theoretically sound posteriors.

5) The manuscript is sloppy at times when discussing related work. "The authors in (Wildberger et al., 2023; Atanackovic et al., 2024) developed FM-based models to estimate the posterior distribution when the prior distribution p(c) of conditions is known. In contrast, our approach tackles situations where the conditions can only be sparsely observed, and the prior distribution is unknown."

The prior distribution p(c) is not known in (Wildberger et al, 2023). They are only able to sample from the joint distributions (c,x), but this does not mean that you can evaluate it. Further, their algorithm can very easily be adapted to the setting you described. If one has posterior samples for sparse conditions $c_i$ one can simply do the joint training over $(x_{i,j}, c_i)$.

6) when style transfer is one of the main modes of motivation, I would also like to see an example of it.

Overall, I appreciate the idea and think that it has merits, but the execution prevents me from accepting it in the current form. I would love to see a practical example, where the main motivation of your algorithm becomes clear. Furthermore, providing a more standard inverse problem on MNIST (with no encoder/decoder) and a continuous condition space would show me that your algorithm at least somewhat scales.  If these problems are discussed/solved, then I am willing to raise my score.

**Questions:**

see weaknesses

---

> ### Author Response · Authors · 2024-11-21
> **Rebuttal by authors (Part 1)**
>
> We appreciate your thorough review of our manuscript and your understanding of the motivation behind our research. Below, we address your concerns in detail. We would be grateful if you could reconsider our rating based on these clarifications.
>
> # For Weakness
>
> > * However, the glaring weakness is that there is not clear cut numerical use case shown. I would like to see a not toyish example where we actually need several conditions and the transport between them. Usually, in the classical inverse problems works there is an implicit geodesic path taken where $y_t=ty+(1−t)y$, since one does not need to alter the condition if posterior sampling is the ultimate goal. If one wants to do style transfer (which seems to be the second motivation of this paper), then one can simply use a conditional FM network which receives the two conditions (source and target) as inputs. Therefore, while theoretically neat I am not convinced of why the generalized continuity equation and a network which moves efficiently also in the condition space, is advantageous. The authors can convince me by providing a clear example where either i) the classical conditional algorithms are not applicable or ii) this approach significantly outperforms the other flow matching models.
>
> Indeed, the primary motivation of our paper is to address conditional generation in scenarios where the conditions are sparsely observed, a situation where "the classical conditional algorithms are not applicable." For example, in the case of style transfer between two known conditions $c_1$ and $c_2$, as you mentioned, one can simply use a conditional FM network that receives the two conditions (source and target) as inputs. However, if either $c_1$ or $c_2$ is unavailable, this FM network cannot be used. Furthermore, even if all conditions $c_1, c_2, \dots, c_M$ are known, and we wish to perform style transfer among them, we would need to train the FM network approximately $\binom{M}{2}$ times, which becomes inefficient as the number of conditions $M$ increases. One of the advantages of our EFM approach is that it can handle the aforementioned intractable situations in FM with a single training of the matrix field model.
>
> > * The scaling in Nc and condition dimension seems to be bad. can you provide the run times for the molecular example also for the baselines? it only says in the appendix that they were completed within 4 hours, but I expect the baselines to train much quicker. Also latent space of a VAE is pretty low dimensional. Please provide training your conditional flow matching model on MNIST (no VAEs..), where the condition space is not discrete (i.e., for instance inpainting). Even if this does not fit your motivation, I would like to see the results in such a more standard example and this would improve my confidence in the scalability.
>
> As you pointed out, the runtime for MMOT-EFM can become significantly longer compared to the baselines when $N_c$ and the condition dimension are large. However, the computation of MMOT should be independent of matrix field learning, which means that the runtime can be significantly reduced by optimizing the implementation, e.g., by introducing parallel computing.
>
> Regarding the runtime, the 4-hour runtime includes the evaluation phase. A significant amount of time was spent on evaluation, and the implementation needs to be fully optimized, which contributes to the longer runtime. We will provide the net runtime without the evaluation phase at a later date. As for the experiment on MNIST with a continuous condition space, such as inpainting, our method may not be well suited from a computational complexity perspective due to the high dimensionality of the condition vector.
>
> > * Appendix D5 and F are empty (or almost empty).
>
> We have deleted these parts.
>
> > * you do not seem to provide any code. I find the algorithm description to be not perfectly clear, there I would very strongly suggest that you at least publish code for the toy example.
>
> We have just submitted the code as supplementary material.

---

> ### Author Response · Authors · 2024-11-21
> **Rebuttal by authors (Part 2)**
>
> > * I believe that the example 7.1 is meaningless. You construct a random example with sparse conditions. Then you show, that your algorithm performs better on the OOD. But basically you can construct an inverse problem which aligns with your in distribution posteriors and does anything else on the OOD data. Of course I am aware that your point is that your algorithm is minimizing the Dirichlet energy and you measure the distribution induced by this. However, it is not clear to me if this is the theoretically optimal thing to do (wrt to Wasserstein). I am guessing that your algorithm computes something like Wasserstein barycenters weighted by some distance to the known conditions? Please clarify why the minimization of the generalized Dirichlet energy should yield theoretically sound posteriors.
>
> We apologize for the lack of background information on Dirichlet energy. The Dirichlet energy represents how sensitive the distribution $p(\cdot \mid c)$ is to perturbations in $c$ with respect to the Wasserstein distance $W_2$. In fact, the following holds:
>
> $$\operatorname{Dir}(p)= \text{``}\lim_{\varepsilon\to0}\text{''} C_k \iint_{\Omega \times \Omega}  \frac{W_2^2(p(\cdot \mid c_1), p(\cdot \mid c_2))}{2 \varepsilon^{k+2}} \boldsymbol{1}_{|c_1-c_2| \leqslant \varepsilon} \mathrm{~d} c_1 \mathrm{d} c_2\quad\text{ for }\quad p\colon\Omega\ni c\longmapsto p(\cdot \mid c)\in\mathcal{P}(D),$$
>
> where $k$ is the dimension of the condition space $\Omega$. For the precise meaning of the limit $\text{``}\lim_{\varepsilon\to0}\text{''}$ and the value of the constant $C_k$, please refer to [§1.3, Lavenant, 2019].
>
> Thus, minimizing the Dirichlet energy is equivalent to yielding posteriors that are not unnaturally sensitive in the Wasserstein sense. The experiment in §7.1 verifies this effect. Specifically, when generating the distribution of unobserved conditions using COT-FM, the distribution changes significantly from $c=(0.25,0)$ to $c=(0.5,0)$. In contrast, using MMOT-EFM reduces the extent of this change.
>
> > * The manuscript is sloppy at times when discussing related work. "The authors in (Wildberger et al., 2023; Atanackovic et al., 2024) developed FM-based models to estimate the posterior distribution when the prior distribution p(c) of conditions is known. In contrast, our approach tackles situations where the conditions can only be sparsely observed, and the prior distribution is unknown." The prior distribution p(c) is not known in (Wildberger et al, 2023). They are only able to sample from the joint distributions (c,x), but this does not mean that you can evaluate it. Further, their algorithm can very easily be adapted to the setting you described. If one has posterior samples for sparse conditions ci one can simply do the joint training over (xi,j,ci).
>
> We apologize for the misunderstanding regarding the work of Wildberger et al., 2023. The contribution of the authors lies in their focus on the fact that the objective function of the FM for conditional generation only requires joint samples $(c,x)$, and they generated these joint samples in a Bayesian manner, such as $x \sim p(x)$ and $c \sim p(c \mid x)$.
>
> However, the treatment of conditions $c$, aside from the aforementioned joint sampling, is almost identical to (OT-C)FM as described in §7. Therefore, if we apply their results to our setup, we can expect similar experimental outcomes to those of (OT-C)FM.
>
> We have clarified this in the revised manuscript.

---

> > ### Comment · Reviewer_a59p · 2024-11-21
> > **Thanks for the rebuttal**
> >
> > I read the rebuttal and you addressed my concern how the dirichlet energy relates to finding "good" posteriors. In style transfer or "inverse problems" it does not happen that $c_1$ or $c_2$ are unavailable (even a conditional generator would suffice I guess since one can take $c_1$ go into latent space, and sample from $c_2$).  I see the point that one can "interpolate" between many conditions using your framework, however the lack of scalability is concerning. I would really appreciate a MNIST example or any imaging example even with a low dimension condition space. Then I would raise my score.

---

> > > ### Author Response · Authors · 2024-11-27
> > > **Thank you very much for the followup!**
> > >
> > > Thank you very much for the follow-up.  We included in the section 7.2 of the revised manuscript the conditional generation of the MNIST dataset with a single image-net background, where the two conditions are “color” and “rotation” that constitute a 4-dimensional condition-space $\Omega=[0, 1]^4$(first dimension is the rotation, and the rests are RGB) . For the training, we used a conditional dataset corresponding to 12 uniform random samples from $\Omega$, and we evaluated the Wasserstein distance from the ground truth (GT) conditional distributions in a manner similar to Figure 4.   Note that these conditionings are nonlinear in the presence of background.   We compared our method against the classifier-free guidance method (Zheng et al. [1]) with different guidance strengths.  On this dataset, we see that the Wasserstein error increases monotonically with the distance of the target conditions from the training set of conditions (For example, c=[0, 0, 0, 0] is very far from the training conditions and it is that much difficult to realize.)
> > >
> > >
> > > Please note that we had to make color and rotation the choice of conditions in this experiment because the default conditional labels of the image benchmarks such as MNIST/CIFAR10 are (1) discrete, and (2) there is no impartial metric on the space of label conditions.
> > > For example, MNISTS are labeled with digits from 0 to 9, but in terms of image generation, 0 is not closer to 1 than it is to 9. The one-hot vector embedding is also not too reflective of actual image generation because 9 is indeed much closer to 4 in terms of a shape than it is to 3.    This is the reason why we limited our experiments to the dataset like ZINC-250k of the form $(x, c(x))$, where $c(x)$ is a feature of $x$.  While there are many datasets of this form in application, for example, in applied fields of science such as economics, biochemistry, and physics, the benchmark datasets/models/publicized architecture are difficult to obtain for these fields.
> > >
> > > [1] Qinqing Zheng and  Matt Le and  Neta Shaul and  Yaron Lipman and  Aditya Grover and  Ricky T. Q. Chen, Guided Flows for Generative Modeling and Decision Making, 2024

---

> > > > ### Comment · Reviewer_a59p · 2024-11-27
> > > > **Thanks**
> > > >
> > > > I will raise my score to 5, but only to 5, since this example does not really address my concerns. First the use of the autoencoder makes it essentially low dimensional and does not show scalability, secondly this example is not really standard which makes the interpretation of the results not easily understandable (i.e., how it performs relatively to other methods). The W_1 plot again relies on OOD generalization for which my issues have not been fully resolved (although your explanation on the dirichlet energy helped) and the trend is not super clear.
> > > >
> > > > Btw, what is guided1.0, 1.1 and 1.2?

---

> > > > > ### Author Response · Authors · 2024-11-29
> > > > >
> > > > > Thank you for reconsidering our score. We would like to address the follow-up concerns.
> > > > >
> > > > > > First the use of the autoencoder makes it essentially low dimensional and does not show scalability.
> > > > >
> > > > > We would like to highlight that many generative models perform flow-based modeling in low-dimensional latent spaces. For instance, the renowned Stable Diffusion model operates in a latent space with dimensions as low as 16 x 16 x 8 (LDM-16) [1], and large-scale chemical research utilizes a latent space of dimension 512 [2].
> > > > >
> > > > >
> > > > > The success of modern generative models often hinges on the careful design of latent spaces and the use of pre-trained networks. Engineering these components is crucial for achieving high-quality results, as demonstrated by the performance of models like Stable Diffusion across various domains.
> > > > >
> > > > >
> > > > > Therefore, while the dimensionality of the latent space is a complexity of MMOT, **it is the combination of advanced latent space design and robust network architectures that drive the scalability and effectiveness of generative models.** We believe our approach aligns with these principles and shows potential for scalability despite using an autoencoder.
> > > > >
> > > > >
> > > > > [1] Rombach et al., High-Resolution Image Synthesis with Latent Diffusion Models, 2022
> > > > > [2] Kaufman et al., latent diffusion for conditional generation of molecules, 2024
> > > > >
> > > > >
> > > > > > this example is not really standard which makes the interpretation of the results not easily understandable (i.e., how it performs relatively to other methods).
> > > > >
> > > > > As mentioned in our previous rebuttal, the provided example is non-standard because we label the images with continuous conditions of rotation and color change rather than standard categorical conditions. This domain of "continuous conditions" is where EFM is designed to excel.
> > > > >
> > > > > However, **it is possible to interpret the results and compare our method's performance with other methods.** In the MNIST experiment, constructing an inverse problem that can take any distribution in OOD is not natural. In this rational experimental setting, the bottom figure in Figure 5 shows that our method has a relatively small worst-case generalization error even in OOD situations with large distances from the training condition.
> > > > >
> > > > >
> > > > >  >  Btw, what is guided1.0, 1.1 and 1.2?
> > > > >
> > > > > We apologize for not clearly explaining this in the revision. In the added experimental results, the values following the label "guided" represent the guidance strength parameter in the classifier-free guidance method.

---

### Official Review · Reviewer_Cdwg · 2024-11-01

**Soundness:** 2
**Presentation:** 2
**Contribution:** 2
**Rating:** 5
**Confidence:** 3

**Summary:**

To achieve extrapolation beyond observed conditions, the authors proposed Extended Flow Matching (EFM) framework that is developed upon the Conditional generative modeling. Specifically, the authors introduced a novel algorithm called MMOT-EFM derived from the Multi-Marginal Optimal Transport (MMOT). In the experiments, the authors showed improved MAE over compared FM-based methods.

**Strengths:**

1. The Extended Flow Matching sounds novel and the authors show the newly introduced conditional components in Fig.1, which is quite intuitive.

2. I like the well-structured theoretical discussion from FM to EFM, this can help domain experts grasp the main contribution and difference between the existing OT-CFM and the proposed MMOT-EFM

**Weaknesses:**

1. I feel concerned about the experimental design. For instance, the authors introduce a rather usual setting (Appendix 1300-1306). Though it aligns well with the synthetic point cloud experiments, it is quite different from the common practice [1].

[1] Ketata M A, Gao N, Sommer J, et al. Lift Your Molecules: Molecular Graph Generation in Latent Euclidean Space[J]. arXiv preprint arXiv:2406.10513, 2024.

2. I think critical experiments against highly related OT-CFM methods are missing in this version.

Alexander Tong, Nikolay Malkin, Guillaume Huguet, Yanlei Zhang, Jarrid Rector-Brooks, Kilian
Fatras, Guy Wolf, and Yoshua Bengio. Improving and generalizing flow-based generative models
with minibatch optimal transport. arXiv preprint 2302.00482, 2023b.

**Questions:**

1. Could you please justify the ZINC-250k experimental design?

---

> ### Author Response · Authors · 2024-11-21
>
> We appreciate your interest in the EFM concept. Below, we address your concerns regarding the molecular generation experiment:
>
> # For Weakness
>
> > * I feel concerned about the experimental design. For instance, the authors introduce a rather usual setting (Appendix 1300-1306). Though it aligns well with the synthetic point cloud experiments, it is quite different from the common practice [1]
>
> The experimental design is motivated by the specific objectives outlined in §1. Our method demonstrates a distinct advantage in scenarios where multiple molecules $x$ share the same label value, and the label value $c$ is sparse. Standard chemical properties, which are continuous and have fewer molecules with identical labels, are not suitable for this experiment. Consequently, we selected a scenario involving the number of bonds, where only two label values are available.
>
> > * I think critical experiments against highly related OT-CFM methods are missing in this version.
>
> The term FM in §7 refers to the OT-CFM method. We apologize for any confusion caused by the lack of clarity. This has been clarified in the revised manuscript.
>
> # For Questions
>
> > * Could you please justify the ZINC-250k experimental design?
>
> ZINC-250k is a widely utilized molecular database of computationally designed compounds, created by G'omez-Bombarelli et al. (2018). It is a more realistic dataset that includes drug-like, commercially available molecules, compared to QM9, which is a standard dataset for molecule generation. We anticipate that similar results would be obtained if trained on other datasets
>
> [[G´omez-Bombarelli et al., ACS Central Science, 2018]](https://pubs.acs.org/doi/full/10.1021/acscentsci.7b00572)

---

> > ### Author Response · Authors · 2024-11-30
> >
> > We appreciate your comments.
> >
> > In our previous response, we addressed your concerns about the molecular generation experiment. In addition, we have performed an additional conditional image generation experiment to further clarify the usefulness of our method; please see the global comment and revision.
> >
> > Your feedback is very valuable. As the peer review deadline is approaching, we would be grateful if you could provide your comments and reconsider your rating as soon as possible.

---

### Official Review · Reviewer_1CpW · 2024-11-03

**Soundness:** 3
**Presentation:** 3
**Contribution:** 2
**Rating:** 6
**Confidence:** 4

**Summary:**

This paper proposes an extension to flow matching to conditional generation on unknown (but related) conditions using a flow on both the data space and the condition space. A variant of this based on multi-marginal optimal transport is proposed as an extension to optimal transport conditional flow matching. 2D and conditional molecular generation experiments are performed showing conditional generation.

**Strengths:**

* Understanding how to extend current generative models to more general conditionals (especially unobserved conditionals) is an important problem particularly in the sciences.
* I enjoyed the symmetry of the presentation of first standard flow matching and OT-CFM settings followed by EFM and MMOT-EFM settings. Table 1 is great to understand the difference to OT-CFM.
* To the best of my knowledge the theory is correct and answers some of my questions on how one might generalize flow matching to condition-augmented spaces.

**Weaknesses:**

* It would be great to make clearer to the reader how this method extends to unseen conditions. I think lines 402-405 kind of get at this, but I would have loved to see more emphasis on this point. It is very easy to design a conditional generative model that technically extends to unseen conditions, but it is much more difficult to enforce that that model extends in a reasonable way. EFM has the potential to guide that extension and I would love to see that point explored further.
* The algorithm is not yet useful in real applications. While the authors also acknowledge this, it’s still a large limitation of the impact of this work. The molecule experiment is extremely limited in terms of comparisons to existing work and overall training setup.
* Much of the theoretical statements are direct extensions from prior work.

**Questions:**

When is MMOT-EFM and EFM in general expected to work better than COT-FM / Bayesian-FM? I know there is a short explanation on the differences in assumptions but it is difficult for me to translate what is gained when making a piecewise continuous assumption on p(x|c) vs. a measurability assumption. It’s not clear to me how this compares to these prior works in general.

Small comments that don’t affect the score:
There appears to be an unfinished section D.5 in the appendix.
GG-EFM isn’t defined in the main text.
I didn’t understand the distinction between p_c and p_{0,c} line 170.
Typo on line 311 “ton he”
Shr\”odinger to Schr\”dinger line 425
The source points in Figure 4 b and c (and corresponding appendix figs) are essentially invisible (grey against a grey background). It would be **really nice** to fix this.


### Overall
I think this work presents an interesting idea with promise to understand how these models generalize to unseen conditions. However, this is not explored theoretically. In addition the current method does not scale to practical settings at the moment. I think further investigation as to when the assumptions behind this method make sense relative to other methods would greatly strengthen this work. A better understanding of how this relates to prior literature and when this method is preferable would likely change my opinion of this work.

---

> ### Author Response · Authors · 2024-11-21
> **Rebuttal by authors (Part 1)**
>
> We appreciate your interest in our EFM theory. We hope that the responses to your questions below will address your concerns comprehensively.
>
> # For Weakness
>
> > * It would be beneficial to elucidate how this method extends to unseen conditions more clearly. Lines 402-405 touch on this, but further emphasis on this point would be valuable. Designing a conditional generative model that technically extends to unseen conditions is straightforward, but ensuring that the model extends in a reasonable manner is more challenging. EFM has the potential to guide this extension, and further exploration of this point would be appreciated.
>
> Our method, MMOT-EFM, extends to unseen conditions by ensuring that the target conditional distribution $p(x \mid c)$ is "smooth." Specifically, we aim to minimize the Dirichlet energy as defined in Equation (3.2). Intuitively, the Dirichlet energy quantifies the sensitivity of the distribution $p(x \mid c)$ with respect to the conditioning vector $c$. Minimizing the Dirichlet energy implies that the sensitivity to the condition $c$ is not excessively large.
>
> While there are numerous methods of extrapolation, it is reasonable to assume an inductive bias that the sensitivity of natural data (e.g., molecules) to conditions (e.g., chemical properties) is not unnaturally large. Therefore, our method addresses extrapolation by training a model such that the data to be extrapolated adheres to this inductive bias of low sensitivity.
>
> We would like to highlight that this type of inductive bias has been historically employed in generative models to prevent overfitting and stabilize generative processes; see, for example, [Miyato et al. in ICLR, 2018](https://openreview.net/forum?id=B1QRgziT-).
>
>
> We have revised the manuscript to further emphasize this point in §1, illustrating how MMOT-EFM guides the extension to unseen conditions in a reasonable manner.
>
> > * The algorithm is not yet applicable to real-world scenarios. While the authors acknowledge this, it remains a significant limitation of the work's impact. The molecule experiment is limited in terms of comparisons to existing work and the overall training setup.
>
> Our method has yet to be evaluated on real-world datasets. However, the ZINC-250k dataset used for conditional molecule generation includes drug-like commercially available molecules, making it more realistic compared to well-known datasets such as QM9.
>
> > * Much of the theoretical statements are direct extensions from prior work.
>
> While it is true that Theorem 3.4 is a direct extension of the fundamental theorem in FM, the theory developed in §3 as a whole should not be seen as a straightforward derivation from existing FM theory. For instance, Proposition 3.1, which introduces a method for utilizing the generalized continuity equation (3.1) for generation and style transfer, is a novel technique within the context of generative models.
>
> # For Questions
>
> > * When is MMOT-EFM and EFM in general expected to outperform COT-FM / Bayesian-FM? Although there is a brief explanation of the differences in assumptions, it is challenging to understand the benefits of assuming piecewise continuity of $p(x|c)$ versus measurability. How does this compare to prior works in general?
>
> Our EFM approach is expected to perform better when there is prior knowledge that the distribution $p(x \mid c)$ we aim to generate changes continuously with respect to $c$. Specifically, it is effective when the sensitivity of $p(x \mid c)$ with respect to the condition $c$, quantified by the Dirichlet energy, is known to be small.
>
> In contrast, COT-FM and Bayesian-FM do not incorporate such prior knowledge regarding the continuity or sensitivity of $p(x \mid c)$ with respect to $c$.
>
> This distinction is evident in the generated results, as shown in Figure 4(b). When generating the distribution of unobserved conditions using COT-FM, the distribution changes significantly from $c=(0.25,0)$ to $c=(0.5,0)$. Conversely, with MMOT-EFM, the extent of this change is mitigated.
>
> We have included this explanation in the revised manuscript.

---

> ### Author Response · Authors · 2024-11-21
> **Rebuttal by authors (Part 2)**
>
> # For Overall
>
> > *  This work presents an intriguing idea with potential to enhance understanding of how these models generalize to unseen conditions. However, this aspect is not theoretically explored. Additionally, the current method does not scale to practical settings. Further investigation into when the assumptions behind this method are valid relative to other methods would significantly strengthen this work. A deeper understanding of how this relates to prior literature and when this method is preferable would likely change my opinion of this work.
>
> We appreciate your evaluation of our concept. Indeed, there are scalability issues, and our method may need further development to be applied to general tasks such as image generation. However, in the context of our motivation, such as molecular generation, we have demonstrated its advantages over existing methods in §7.2. Regarding the relation to prior literature, particularly the differences from COT-FM and Bayesian FM, we have clarified these points in our response to your previous question. Please refer to that response for a detailed explanation.

---

> > ### Comment · Reviewer_1CpW · 2024-11-25
> >
> > I thank the authors for their rebuttal. While MMOT-EFM outperforms previous methods on some toy tasks, I still feel it is lacking a useful application at scale, and it is unclear to me that this method is useful outside of toy settings given its additional computational cost and complexity. For this reason I maintain my score.

---

> > > ### Author Response · Authors · 2024-11-30
> > >
> > > We appreciate your comments. In response to your comments, we have added a new conditional image generation experiment. This experiment clearly demonstrates the scalability of our EFM. For details on the experiment settings, etc., please see the global comment and revision.
> > >
> > > Your feedback is very valuable. The review deadline is approaching, so we would be grateful if you could provide your comments and reconsider your rating as soon as possible.

---

> > > > ### Comment · Reviewer_1CpW · 2024-12-03
> > > >
> > > > I thank the authors for their response and additional experiments. This partially satisfies my main concern of empirical scalability and practical applicability as such I raise my score 5-->6.
> > > >
> > > > Why not higher:
> > > > * While I like the idea of enforcing smoothness of predicted distribution relative to the condition space, I'm not sure the killer application has been identified.
> > > > * The experiments here are still limited in scale. Combined with the reasonable, but somewhat limited novelty on the theory side given prior work on conditional flow matching, I think less toy experiments would significantly improve the impact of this work.
> > > >
> > > > Why not lower:
> > > > * I believe the authors have demonstrated an interesting method towards generalizing conditional flows towards unseen (but related) conditions.
> > > > * I'm not concerned with mini-batch approximation errors and believe this leads to a relatively scalable algorithm.
> > > > * While shown in some somewhat niche experimental settings, I believe this work is valuable in its current state to those applying flow matching to scientific applications.

---

### Official Review · Reviewer_mdMu · 2024-11-03

**Soundness:** 2
**Presentation:** 2
**Contribution:** 3
**Rating:** 6
**Confidence:** 3

**Summary:**

Flow matching can generate different data distribution given different desired property conditions. The authors proposed the extended flow matching (EFM) which introduces a continuous mapping from a joint continuous space of time and property conditions to corresponding data distribution, which enables smooth shifts between different conditions. The authors also extended optimal transport to Multi-Marginal Optimal Transport (MMOT) for multiple property conditions. They validated their method on a 2D toy model and conditional molecular generation.

**Strengths:**

The theory of integrating property conditions and time in flow matching is highly innovative, and the authors developed MMOT to perform optimal transport within this space.

**Weaknesses:**

The experimental evidence is insufficient.

**Questions:**

Major:

1.	Could the authors explain or give an intuition about the regression in MMOT (Eq. 3.4)?

2.	Could the authors show the extrapolation ability of their methods in a more realistic application of EFM, e.g. style transfer of images?

Minor:

1.	At the end of Line 311, “focus on the” is misspelled as “focus ton he”.

2.	“ConvHull” should be explained.

---

> ### Author Response · Authors · 2024-11-21
> **Rebuttal by authors**
>
> We appreciate your thorough review of our manuscript. Below, we provide detailed responses to your inquiries. We hope that these clarifications will enhance your confidence in your evaluation.
>
> # For Questions
>
> > * Could the authors explain or give an intuition about the regression in MMOT (Eq. 3.4)?
>
> Equation (3.3) serves as a soft constraint for the boundary conditions in Equation (3.4). The meaning of Equation (3.4) is that the conditional distributions generated by the model should match the known data distributions $\mu_\xi$, $\xi \in A$. This is analogous to the requirement in standard FM that the endpoints of the probability path generated by the model correspond to the source and target distributions. In the case of FM, where only two distributions are considered, this can be achieved by simply connecting them with a straight line, eliminating the need for regression. However, in EFM, we need to consider three or more distributions simultaneously, making it hard to satisfy the constraints strictly. Therefore, we relax the constraints to be soft in this context.
>
> > * Could the authors show the extrapolation ability of their methods in a more realistic application of EFM, e.g. style transfer of images?
>
> Our method is designed to work in situations where the dimensionality of the conditioning vector is relatively low, and the conditions are sparsely observed. Such scenarios are common in the context of molecular generation, as demonstrated in §1 and §7.2. Therefore, EFM is expected to show extrapolation ability in a more practical molecular generation.
>
> We have incorporated the aforementioned explanation into the introduction of the revised manuscript.

---

> > ### Author Response · Authors · 2024-11-30
> > **Reminder from authors**
> >
> > We hope this message reaches you well. We are writing to remind you that the deadline for submitting comments on our ICLR 2025 rebuttal is approaching.
> >
> > We have revised our paper and added an experiment on MNIST; please see the global comment. We believe that these results will address your concerns.
> >
> > Your feedback is very valuable to us, so if you have time, we would be grateful if you could provide your comments and reconsider your rating.

---

> > > ### Comment · Reviewer_mdMu · 2024-12-02
> > > **Thanks**
> > >
> > > Thank you for clarifying the methods and providing the experiment results on MNIST.
> > >
> > > I still think that the practical applications demonstrated in the paper are limited. The authors also mention that their method is only applicable to low-dimensional conditioning vectors, such as the molecule generation showcased in the study. However, I believe that there should be other potential applications beyond this. I will maintain the score.

---

### Official Review · Reviewer_QpJX · 2024-11-04

**Soundness:** 1
**Presentation:** 1
**Contribution:** 1
**Rating:** 3
**Confidence:** 4

**Summary:**

The authors propose extended flow matching (EFM) for conditional sampling and style transfer using flow matching. EFM consists of

1. learning a field which also uses the conditioning vector $c$ as input, which the authors call a matrix field.
2. The authors then integrate the learned field $u(x, t, c)$ along different paths $\gamma: [0, t] \rightarrow [0, 1] \times C$, where $C$ is the set of conditioning vectors.
    1. For instance, for conditional generation the authors propose integrating along the path $\gamma(t) = (t, c)$, which reduces to conditional flow matching.
    2. For style transfer, the authors integrate along the path $\gamma(t) = (1, (1-t) c_1 + t c_2)$. Since integrating along $\gamma(t)$ can be out of domain for models learned trained just on pairs $x, c \sim p(x, c)$, the authors propose a learning algorithm such that the field $u_\theta$ also observes such paths during training.

The authors propose learning such a field $u$ using optimal transport:

1. the authors propose learning an optimal plan similar to [Lipman et al 2023]
2. instead of using linear interpolation between different points on a path, the authors extend the set of paths to include functions belonging to an RKHS.

**Strengths:**

The authors identify an interesting problem: observing the conditioning vector in a number of domains can be hard or expensive. The proposal of integrating along paths between different marginals is also interesting, a similar proposal is studied in [Albergo et al 2023].

**Weaknesses:**

1. While the motivation of EFM was to provide ensure that the learned network $u(x, t, c)$ is smooth with respect to the conditioning vector $c$, the authors do not address how imposing smoothness can allow extrapolation to conditioning vectors not seen during training.
2. Could the authors explain why the multi-marginal optimal transport approach allows for extrapolating to conditioning vectors not seen during training?
3. The authors should also consider including other works that learn multi-marginal flow models? For instance, [Albergo et al 2023] propose learning multi-marginal flows and present a learning algorithm for optimizing the paths such that the transport cost in $W_2$ metric is minimized.
4. [Albergo et al 2023] also propose a much more general algorithm for including paths between samples from an arbitrary number of marginal distributions, available during training.
5. The experiments section can be improved by adding extra text explaining the results and the figures, particularly in figure 4.


[Albergo et al 2023]  Albergo, M.S., Boffi, N.M., Lindsey, M. and Vanden-Eijnden, E., 2023. Multimarginal generative modeling with stochastic interpolants. arXiv preprint arXiv:2310.03695.

**Questions:**

1. Can the authors consider providing definitions before introducing a new notation in the text?
2. What is the effect of defining $\pi$ using plans built using batched samples? Would the vector/matrix field learned change as a function of the batch size?
3. What kernels do the author use for the RKHS used to construct paths?
4. In lines 212-214 and lines 220-222, can the authors clarify the output of $u$?
5. the discussion about the weak assumption of measurability and continuity of $p(x|c)$ with respect to $c$ requires clarification, particularly since piece-wise continuous functions are measurable as well.

---

> ### Author Response · Authors · 2024-11-21
> **Rebuttal by authors (Part 1)**
>
> We appreciate your valuable feedback and apologize for any lack of clarity in our initial explanation. Below, we address your concerns and questions:
>
> # For Weaknesses
>
> > * While the motivation of EFM was to provide ensure that the learned network $u(x,t,c)$ is smooth with respect to the conditioning vector $c$, the authors do not address how imposing smoothness can allow extrapolation to conditioning vectors not seen during training.
> > * Could the authors explain why the multi-marginal optimal transport approach allows for extrapolating to conditioning vectors not seen during training?
>
> We apologize for the unclear wording in line 86. First, we would like to clarify the meaning of "smoothness," as mentioned in our motivation. Our goal is to ensure that the conditional distribution $p(x \mid c)$ which we will generate is "smooth," which means minimizing the Dirichlet energy as defined in Equation (3.2). Intuitively, the Dirichlet energy represents the sensitivity of the distribution $p(x \mid c)$ with respect to the conditioning vector $c$. Thus, minimizing the Dirichlet energy implies that the sensitivity with respect to the condition $c$ is not too large.
>
> Although there are infinite ways of extrapolation, it is reasonable to assume an inductive bias that the sensitivity of data in nature (e.g., molecules) to conditions (e.g., chemical properties) is not unnaturally large. Therefore, our method addresses extrapolation by learning a model such that the data to be extrapolated follows this inductive bias of low sensitivity.
>
> We would like to note that this kind of inductive bias has been used throughout the history of generative models as a method to prevent overfitting and a method to stabilize generative models; see, for example,  [Miyato et al. in ICLR, 2018](https://openreview.net/forum?id=B1QRgziT-).
>
> Our experiments in §7 demonstrate that EFM, which minimizes the Dirichlet energy, outperforms methods that do not minimize this energy (such as FM and COT-FM) in terms of generation performance.
>
> In addition, the cost (objective) function used in our multi-marginal optimal transport (MMOT) approach provides an upper bound on the Dirichlet energy; please refer to lines 233-236 and Table 1. Therefore, optimizing the transport plan $\pi$ through the MMOT approach also minimizes the Dirichlet energy, which in turn reduces the sensitivity of the generated distribution $p(x \mid c)$ with respect to the conditioning vector $c$.

---

> ### Author Response · Authors · 2024-11-21
> **Rebuttal by authors (Part 2)**
>
> > * The authors should also consider including other works that learn multi-marginal flow models? For instance, [Albergo et al 2023] propose learning multi-marginal flows and present a learning algorithm for optimizing the paths such that the transport cost in W2 metric is minimized.
> > * [Albergo et al 2023] also propose a much more general algorithm for including paths between samples from an arbitrary number of marginal distributions, available during training.
>
> In fact, our proposed EFM framework indeed incorporates the approach of [Albergo et al., 2023].
>
> More specifically, one of the implementations of EFM, Geo-EFM (supplemented in Appendix F), and [Albergo et al., 2023] both use the same transport plan $\pi$ to learn the matrix (or vector) field. In addition, our interpolator $\bar\psi(c \mid (x_i)_i)$ in Table 1 corresponds to the barycentric stochastic interpolant $x(\alpha) = \sum_i x_i \alpha_i$ in [Equation (5), Albergo et al., 2023]. Here, the interpolation coordinates $\alpha = (\alpha_i)_i \in \Delta^K$ valued in the simplex $\Delta^K$, can be roughly regarded as the condition vector $c \in \Omega$ in our case, i.e., $\Delta^K \approx \Omega$. More precisely, the condition $c$ is given by expanding it on a basis. That is, $\alpha$ is the coefficient when expanding $c$ as $c = \sum_i c_i \alpha_i$ in a basis $(c_i)_i$. Consequently, Geo-EFM, similar to [Albergo et al., 2023], can utilize an arbitrary number of marginal distributions during the training process.
>
> The only difference between Geo-EFM and the method in [Albergo et al., 2023] is that, after learning the vector field, Albergo et al. optimize the path on the space of "conditions" $\alpha \colon [0,1] \to \Delta^K \approx \Omega$ such that the transport cost in the W2 metric is minimized. Thus, we can take the same procedure as above to optimize $\gamma \colon [0,1] \to \Omega$ with the Geo-EFM setting.
>
> Our MMOT-EFM is novel in that it minimizes the transport cost in a complementary way to the optimization of $\gamma$. MMOT-EFM trains a matrix field, which is an extension of a vector field, to also minimize a generalization of the transport cost called Dirichlet energy. This makes it possible to learn a model that transports optimally with only one training of the model without optimizing $\gamma$. We note that there is a computational limitation on the number of marginal distributions (as you pointed out) due to the use of MMOT during training.
>
> Moreover, in our setting, the set of conditions $\Omega$ is continuous, whereas in [Albergo et al., 2023], it is discrete.
>
> In summary, MMOT-EFM and [Albergo et al., 2023] can be seen as complementary approaches. We have mentioned the above in the revision.
>
> > * The experiments section can be improved by adding extra text explaining the results and the figures, particularly in figure 4.
>
> We apologize for the inconvenience. Due to page limitations, we could not include sufficient explanation in the submitted manuscript. We have added more explanations to the caption in the revision.
>
> # For Questions
>
> > * Can the authors consider providing definitions before introducing a new notation in the text?
>
> We apologize for the technical nature of the mathematical equations. We would like to consider your comments on the manuscript. Could you tell us exactly where we introduce the new notation before the definition?
>
> > * What is the effect of defining $\pi$ using plans built using batched samples? Would the vector/matrix field learned change as a function of the batch size?
>
> Larger batch sizes tend to stabilize the learning process because the changes in the matching $\pi$ per iteration become smaller.
>
> > * What kernels do the authors use for the RKHS used to construct paths?
>
> In general, one can employ any nonlinear kernel.
> In the implementation of this paper, we use the linear kernel.
>
> > * In lines 212-214 and lines 220-222, can the authors clarify the output of $u$?
>
> Here, $u$ returns a matrix of size $d \times (1+k)$, where $d$ is the dimension of the data $x \in D$ and $k$ is the dimension of the condition $c \in \Omega$. In general, the output of $u$ is of size $d \times \operatorname{dim} \Xi$ (see Proposition 3.1). In §3.1, $\Xi = I \times \Omega$, so $\operatorname{dim} \Xi = 1+k$.
>
> > * The discussion about the weak assumption of measurability and continuity of $p(x \mid c)$ with respect to $c$ requires clarification, particularly since piece-wise continuous functions are measurable as well.
>
> We apologize for the confusion. Our intention was not clearly conveyed. In COT/Bayesian-FM, the conditional distribution $p(x \mid c)$ is assumed to be measurable with respect to $c$, which allows $p(x \mid c)$ to change discontinuously. In contrast, we assume that $p(x \mid c)$ changes continuously, or more precisely, that the sensitivity of $p(x \mid c)$ with respect to the condition $c$ is small. This difference is demonstrated in §7.1, Figure 4(b).
>
> We have included this clarification in the revision.

---

> > ### Comment · Reviewer_QpJX · 2024-11-22
> >
> > Thank you for your response to the review. I have a few more questions below:
> >
> > > Larger batch sizes tend to stabilize the learning process because the changes in the matching π per iteration become smaller.
> > >
> >
> > Can the authors comment on the convergence of the finite-sample approximation of the coupling $\pi$ to the coupling  $\pi^*$ which minimizes $E_{\pi}\|x_0 - x_1\|_2^2$? Any empirical or theoretical analysis would make the case stronger. More specifically, as the authors claim on line 276, can the optimal coupling $\pi^*$ be approximated by finite-samples? a proof or asymptotic analysis would be appreciated.
> >
> > From appendix D.3 it seems the cost of MMOT is prohibitive, therefore the authors propose an approximation. Can the authors discuss the effect of this approximation on the claim that they minimize an upper bound on the Dirichlet energy?
> >
> > I believe the authors have a typo in their rebuttal, there is no discussion of any Geo-EFM algorithm in appendix F, rather it contains a description of metrics, datasets and baselines.
> >
> > > Our MMOT-EFM is novel in that it minimizes the transport cost in a complementary way to the optimization of $\gamma$
> > >
> >
> > Can the authors why a more expensive procedure is a better option? Can the authors provide a graph containing the computational burden of computing the plan $\pi$ as the batch size increases for a high-dimensional (d > 100) dataset.
> >
> > > We apologize for the technical nature of the mathematical equations. We would like to consider your comments on the manuscript. Could you tell us exactly where we introduce the new notation before the definition?
> > >
> > 1. the paragraph from lines 148-161, including equation 2.3.
> >
> > > Although there are infinite ways of extrapolation, it is reasonable to assume an inductive bias that the sensitivity of data in nature (e.g., molecules) to conditions (e.g., chemical properties) is not unnaturally large
> > >
> >
> > Can the authors show any examples of such an inductive bias helping in solving any high-dimensional inverse problems?  For instance, with the MNIST/CIFAR10, etc datasets? Or class-conditional generation?
> >
> > Can the authors give a precise, not a rough, set of differences to the multi-marginal modeling approach introduced in [Albergo et al 2023] and the added advantage of their approach?

---

> ### Author Response · Authors · 2024-11-27
> **Thank you very much for the followups and discussion!  (Part1)**
>
> Thank you very much for the follow-up, we would like to address your concerns below.
>
> >* Can the authors comment on the convergence of the finite-sample approximation of the coupling $\pi$ to the coupling $\pi^\ast$ which minimizes $\mathbb{E}_\pi |x_0−x_1|_2^2$? Any empirical or theoretical analysis would make the case stronger. More specifically, as the authors claim on line 276, can the optimal coupling $\pi^\ast$ be approximated by finite-samples? a proof or asymptotic analysis would be appreciated.
>
> It is known that the finite sample approximation converges asymptotically for any $\pi$ that minimizes $\mathbb{E}_\pi |x_0−x_1|_2^2$, for example [[Theorem 5.10, Villani, 2009]](https://rdcu.be/d09GX). We also conjecture that the multi-marginal case in line 276 converges in a similar way.
>
> >* From appendix D.3 it seems the cost of MMOT is prohibitive, therefore the authors propose an approximation. Can the authors discuss the effect of this approximation on the claim that they minimize an upper bound on the Dirichlet energy?
>
> In this approximation, MMOT is applied only at the cluster level, and the coupling between each cluster is conducted using a method of the user’s choice.  Indeed, this has an effect of achieving the energy value that is greater than the upper bound shown in the objective function in 3.2 because, in the implementation, the infimum in $ \inf_\pi \int_{D^{|A|} \times \Xi } \| \nabla_\xi \phi(\xi | x_A)  \|^2 \pi (d x_A) dc  $ will be taken not over the space of all joint distributions  $\mathcal{P} ( D^{|A|}) $, but over its subset of form $\pi_{\mathrm{user}} ( x_A  | \lbrace m_{i} \rbrace  ) \pi_{\mathrm{cluster}}( \lbrace m_{i} \rbrace ) $ where $\pi_{\mathrm{cluster}}$ is computed with discrete MMOT over  $\lbrace m_{ik}  \rbrace$ and $ \pi_{user}$ is chosen by the user.
>
> Indeed, $\pi_{user}$ can be chosen to be couplings that respect the consideration to kinetic energy as well, such as the generalized Geodesics coupling based on optimal transport.
> Also, by definition, this approximation shall converge to the actual upper bound if we choose $|U_{ik}| = 1$ take the limit of $|B_i| \to \\infty.$
>
> > I believe the authors have a typo in their rebuttal, there is no discussion of any Geo-EFM algorithm in appendix F, rather it contains a description of metrics, datasets and baselines.
>
> We are sorry for the typo. Our discussion of Geo-EFM in the currently uploaded version (modified: 21 Nov 2024) is provided in Section E, with the title *A REMARK ON GENERALIZED GEODESIC COUPLING(GGC) AND THE SAMPLING OF $\bar{\psi}$.*
>
>
> > (Undefined notations  at) the paragraph from lines 148-161, including equation 2.3.
>
> Thank you for specifying the spot. We stated the definitions before making the statement in 148-161, just as below:
>
> >Let $\psi$ be a random path such that $\psi\colon I \rightarrow D$ is differentiable. Let $Q$ be a distribution over a space $H(I; D) \coloneqq \Set{\psi\colon I \rightarrow D | \psi \text{ is differentiable}}$ of paths that map time $t\in I$ to data $x \in D$, and use $\mu^\psi_t$ to denote $\delta_{\psi(t)}$.
> >With these definitions, we can present $\mu = \mu^Q$ from a random path $\psi$ as
>
> We have revised the phrases with similar expressions when we can.
>
> > Can the authors show any examples of such an inductive bias helping in solving any high-dimensional inverse problems? For instance, with the MNIST/CIFAR10, etc datasets? Or class-conditional generation?
>
> The difficulty in providing the efficacy of our method on image benchmarks like MNIST/CIFAR10 is that the conditions in these datasets are (1) discrete and (2) there is no impartial metric on the space of conditions.
> For example, MNISTS are labeled with digits from 0 to 9, but in terms of image generation, 0 is not closer to 1 than it is to 9. One hot vector embedding is also not too reflective of actual image generation because 9 is indeed much closer to 4 in terms of an image than it is to 3.    This is the reason why we restricted our experiments to the dataset like ZINC-250k of the form $(x, c(x))$, where $c(x)$ is a feature of $x$.  While there are many datasets of this form in application, for example, in applied fields of science such as economics, biochemistry, and physics,  benchmark datasets, models,  and publicized architecture are difficult to obtain.  (continued to next part)

---

> > ### Author Response · Authors · 2024-11-27
> > **Thank you very much for the followups and discussion! (Part2)**
> >
> > However, we included in the section 7.2 of the revision the conditional generation of the MNIST dataset with a single image-net background, where the two conditions are “color” and “rotation” that constitute the 4-dimensional condition-space $\Omega=[0, 1]^4$(first dimension is the rotation, and the rests are RGB) . For the training, we used a conditional dataset corresponding to 12 uniform random samples from $\Omega$, and we evaluated the Wasserstein distance from the ground truth (GT) conditional distributions in a manner similar to Figure 4.   Note that these conditionings are nonlinear in the presence of background.   We compared our method against the classifier-free guidance method (Zheng et al. [1]) with different guidance strengths.
> >
> >
> > > Can the authors give a precise, not a rough, set of differences to the multi-marginal modeling approach introduced in [Albergo et al. 2023] and the added advantage of their approach?
> >
> > Our method differs from [Albergo et al. 2023]  in that we do not necessarily require the path optimization step (Corollary 3)  in  [Albergo et al. 2023].
> > To further clarify the difference,  let us first provide our understanding of  [Albergo et al. 2023] in a procedural format.
> > ##  [Albergo et al. 2023]‘s approach
> > Here is the order of steps by which Albergo et al. construct a transportation plan:
> > 1. Define the stochastic process (Barycentric Stochastic interpolant, eqn 5)   $$x(\alpha) = \sum_i x_i \alpha_i,  ~~~~~ (x_0, …. x_n) \in \pi$$
> > where $\pi$ is produced via barycentric interpolation of a set of optimal transport from $\mu_0 \to \mu_k$ (eqn 18).
> > $mu_0$ is chosen as a uniformative distribution (e.g. Gaussian)
> > This will define a map $\alpha \to \mu_\alpha$, where $x(\alpha) \sim \mu_\alpha$.
> >
> > 2.  Learn the vector field $g_k$ in the system of continuity equation for $(\lbrace g_k \rbrace , \mu_\alpha) $  eqn(7)
> > by leverating Theorem1.
> >
> > 3.  Given an endpoint distribution $\rho_i, \rho_j$, optimize the path $\alpha : I \to \Omega$ with $\alpha(0) = \rho_i, \alpha(1) = \rho_j$
> > for the energy $  \int_0^1 \mathbb{E}[| [g_1, …. g_n]  \dot \alpha(\alpha(t), t) |^2]   dt   $ , obtaining the optimal $\alpha^*$
> >
> > 4.  Generate a path from  $\rho_i$ to $\rho_j$ via the ODE  $\dot{x}(t) = [v_1, …. v_d]  \dot \alpha^*(t) $
> >
> > ##  EFM
> > Meanwhile, this is the way we solve the interpolation problem
> >
> > A.
> > 1. Obtain the coupling $\pi$ of your choice  over $ (x_1, …. x_n) $. In our paper,  we present (i) MMOT and (ii) Generalized Geodesic.  (ii) is
> > the same as the coupling used in (1) above.
> >
> > 2.   Use (eqn 3.4) to construct a stochastic process $$\psi(\alpha) = \phi(\alpha | x_1, …. x_n),   with  (x_1, …. x_n) \sim \pi   $$
> > This results ina map   $\alpha \to \mu_\alpha$, where $psi(\alpha) \sim \mu_\alpha$.
> >
> > B.
> >
> > Learn the matrix field $u$ in the generalized continuity equation for  $(u,   \mu_\alpha) $
> >
> > C.
> >
> > Given an endpoint  distribution  $\rho_{c_0},   \rho_{c_1}$,  generate a path from $\rho_{c_0}$ to $\rho_{c_1}$ via the ode
> >  $\dot{x}(t) = u \dot{\gamma}(t) $, where $\gamma(t) = c_1- c_0$.
> >
> > ## The difference between  [Albergo et al. 2023]  and EFM
> > EFM differs from  [Albergo et al. 2023] in that our procedure does not require the equivalent of (3) in [Albergo et al. 2023].  In fact,  the procedure [(1) (2) (3) (4)] of  [Albergo et al. 2023] and the procedure [A, B, C]  of EFM  can be precisely aligned, and our absence of the requirement of (3) is the exact difference between them.
> >
> > To be more precise, note that our A1 and A2 correspond to  (1),  B corresponds to (2), and C corresponds to (4).     Also, $x(\alpha)$ corresponds to our $\psi(\alpha)$, and $[g_1, …. g_n]$ corresponds to our $u$, and $\alpha$ corresponds to $\gamma$.
> > We do not necessarily require (3) in our procedure because,  instead of optimizing the path $\alpha$ on the condition space that minimizes the pairwise kinetic energy through the weight $\lbrace g_k \rbrace$, we choose a process $\psi$ on the observation space that (approximately) minimizes the multimarginal analogue of the kinetic energy that is Dirichlet energy of $u$.
> > ## Advantages
> > - Our approach does not require the optimization of  $\alpha = \gamma$ for every interpolation.
> > - Our approach can be modified to combine [Albergo et al. 2023] ’s approach by including the analogue of their (3) before (C).
> > More particularly, if we choose to execute [A1, A2, B, (3), C] in order,  the target velocity field will be theoretically the same as Albergo’s approach when  (i) we choose Generalized Geodesic in A1 and (ii) choose linear regression in A2.
> >
> > This way, our method is complementary to   [Albergo et al. 2023].  Our method offers “an additional” venue that uses the optimization of the “stochastic process” itself.

---

> > > ### Author Response · Authors · 2024-11-30
> > > **Reminder from authors**
> > >
> > > We hope this message reaches you well. We are writing to remind you that the deadline for submitting comments on our ICLR 2025 rebuttal is approaching.
> > >
> > > In our previous reply, we explained the differences between [Albergo, et al. 2023] and our work, which was a concern of yours. We have also revised our paper and added an experiment on MNIST; please see the global comment.
> > >
> > > Your feedback is very valuable to us, so if you have time, we would be grateful if you could provide your comments and reconsider your rating.

---

> > > > ### Comment · Reviewer_QpJX · 2024-12-01
> > > >
> > > > > It is known that the finite sample approximation converges asymptotically.
> > > > Can the authors clarify where Theorem 5.10 in Villani, 2009 discusses the convergence of finite-sample approximations?
> > > >
> > > > > We also conjecture that the multi-marginal case in line 276 converges in a similar way
> > > > Since this conjecture and the finite sample convergence in the coupling of two distributions are an important part of the paper, can the authors provide citations which discuss convergence properties or proofs?
> > > >
> > > > > Our approach does not require the optimization of $\gamma$ for every interpolation.
> > > >
> > > > Optimizing the interpolant in step 3 of [Albergo et al 2024] is significantly cheaper than solving A for EFM. Moreover,  [Albergo et al 2024]  are able to scale their experiments up to 64 x 64 x 3 dimensions, significantly higher than the .
> > > >
> > > > > The difficulty in providing the efficacy of our method on image benchmarks like MNIST/CIFAR10 is that the conditions in these datasets are (1) discrete and (2) there is no impartial metric on the space of conditions
> > > >
> > > > There are plenty of high-dimensional posterior sampling problems that have been studied with diffusion models, with several metrics such structural similarity, perceptual metrics such LPIPS, MAE, MSE, etc. Albeit they are imperfect but there are several benchmarks for these tasks. Also, can the authors explain what they mean by an impartial metric?
> > > >
> > > > Can the authors provide runtime on solving the finite-sample optimal transport for data of size ~ 12888 with a batch size of 128? More generally, for a single batch what is the run time for all the steps in section A as identified by the authors.
> > > >
> > > > Thank you for adding the MNIST digits on CIFAR10 background experiment, can the authors provide details on the baselines as well as the metrics used. To the best of my understanding, $W_1$ is also an integral probability metric which requires solving a high-dimensional optimization over Lipschitz functions with Lipschitz constant less than 1.

---

> > > > > ### Author Response · Authors · 2024-12-02
> > > > >
> > > > > Thank you for your insightful questions and for engaging in this discussion. Here are our responses:
> > > > >
> > > > > > Can the authors clarify where Theorem 5.10 in Villani, 2009 discusses the convergence of finite-sample approximations?
> > > > >
> > > > > We apologize for the confusion. The correct reference is Theorem 5.20 (Stability of optimal transport). In this theorem, please consider the sequence $(\mu_k)_k$ as the sequence of finite-sample approximations.
> > > > >
> > > > > > Since this conjecture and the finite sample convergence in the coupling of two distributions are an important part of the paper, can the authors provide citations which discuss convergence properties or proofs?
> > > > > > Can the authors provide runtime on solving the finite-sample optimal transport for data of size ~ 12888 with a batch size of 128? More generally, for a single batch what is the run time for all the steps in section A as identified by the authors?
> > > > >
> > > > > Firstly, we do not consider the finite-sample convergence properties to be of paramount importance in this paper. Approximating the optimal transport coupling using finite samples is a well-established method in generative models and is not novel. For example, please refer to:
> > > > > - Tong et al., TMLR, 2024
> > > > >
> > > > > Additionally, the convergence of the coupling itself has been studied less than the convergence of the minimal transport cost due to the non-uniqueness of the coupling.
> > > > >
> > > > > > To the best of my understanding, $W_1$ is also an integral probability metric that requires solving a high-dimensional optimization over Lipschitz functions with Lipschitz constant less than 1.
> > > > >
> > > > > When computing the $W_1$ distance between two probability distributions $\mu_{0}$ and $\mu_1$ numerically, you can use the ``ot.emd`` function from the Python Optimal Transport library. This function does not use the formulation of the integral probability metric; see the documentation.
> > > > >
> > > > > > Optimizing the interpolant in step 3 of [Albergo et al. 2024] is significantly cheaper than solving A for EFM. Moreover, [Albergo et al. 2024] are able to scale their experiments up to 64 x 64 x 3 dimensions, significantly higher than the.
> > > > >
> > > > > While the optimization in Albergo et al. (2024) might be relatively cheaper, their method is only applicable when the conditions are discrete. Our method can be used when the conditions are continuous, meaning that there is a natural (Euclidean) distance between the conditions.
> > > > >
> > > > > > Also, can the authors explain what they mean by an impartial metric?
> > > > >
> > > > > In the previous response, we used the word “impartial” with respect to the specific dataset in question when using the “categorical label.” We apologize for any confusion caused by our use of the word *metric* to refer to the metric or distance in the conditions, not the *evaluation* metric.
> > > > >
> > > > > Our task of generating a conditional distribution of unobserved $c$ in continuous space of conditions $\Omega$ is mainly dependent on the *metric* on $\Omega$ because it determines how close a given $c$ is to another. For example, if the target condition $c_*$ is very close to $c_0$ in one metric (say $d_1$), it is hoped that $\mu_{c_*}$ is distributionally close to $\mu_{c_0}$. This may not be the case for another metric (say $d_2$), where $d_2(c_*, c_0) \gg d_1(c_*, c_0)$, leading to a vastly different $\mu_{c_*}$ from $\mu_{c_0}$. Our method aims to construct a model $\mu\colon \Omega \to \mathcal{P}(D)$ such that $\mu_c$ is as smooth in $c$ as possible.
> > > > >
> > > > > In the MNIST example with categorical conditions, we believe there is no *impartial* metric on the conditions of *digits* when considering a conditional distribution of *images*, especially because there are no ground-truth datasets other than those used in training ($\{0, …, 9\}$). This is not the case for the molecule dataset, where an appropriate continuous label with a ground-truth dataset is present but not in the training dataset.
> > > > >
> > > > > > Thank you for adding the MNIST digits on CIFAR10 background experiment, can the authors provide details on the baselines as well as the metrics used.
> > > > >
> > > > > For the baseline method, we used the method of (Zheng et al. [1]), where the same model as ours was used to train the velocity field $\mathbb{R}^{d_x}\times\mathbb{R}\times\mathbb{R}^{d_c} \ni(x, t, c) \mapsto v(x,t,c)\in\mathbb{R}^{d_x}$, except that instead of outputting a matrix of size $(d_c + 1) \times d_x$, we used a trainable linear map of size $(d_c + 1) \times 1$ to convert the matrix to a vector of shape $1 \times d_x$. We trained the model for the same number of iterations as ours. For evaluation, we used $W_1$ distance between the generated distribution $\hat \mu_c$ and the ground truth $\mu_c$, approximated by applying ``ot.emd`` on batches of size 10000 sampled from $\hat \mu_c$ and $\mu_c$.
> > > > >
> > > > > We hope these responses address your concerns. We would greatly appreciate it if you could consider improving our rating based on this clarification. Thank you for your valuable feedback.

---

> > > > > > ### Comment · Reviewer_QpJX · 2024-12-02
> > > > > >
> > > > > > > We apologize for the confusion. The correct reference is Theorem 5.20 (Stability of optimal transport).
> > > > > >
> > > > > > Thank you for citing the correct reference. The reference mentions that the finite-sample approximations lead to consistent estimators of the optimal transport plan, however there are no mentions of producing unbiased estimates. Would that apply that using the finite-sample approximation, rather than the intractable transport plan, in the equations on lines 265 not yield an upper bound? So in effect, the authors are not able to upper bound the Dirichlet energy with their objective?
> > > > > >
> > > > > > > While the optimization in Albergo et al. (2024) might be relatively cheaper, their method is only applicable when the conditions are discrete.
> > > > > >
> > > > > > Their method is applicable to marginals distributions with continuous-valued support, see definition 1.
> > > > > >
> > > > > >
> > > > > > > In the MNIST example with categorical conditions, we believe there is no impartial metric on the conditions of digits when considering a conditional distribution of images, especially because there are no ground-truth datasets other than those used in training $(0, \dots, 9)$.
> > > > > >
> > > > > > This is incorrect. In inverse problems, one can define a matrix $A$ and observations $y = A x + \varepsilon$, where $x$ is the image and $\varepsilon$ is mean zero noise, see Chung et al 2022 for several examples of $A$. Here, generating $p(x | y)$ is a problem that has been studied for a few decades now, and recently with diffusion models, and $y$ is considered a "ground truth label" since you are generating it using a linear forward process.
> > > > > >
> > > > > > More typically, one can always consider generating labels $y = g(x)$ for a deterministic function $g$ and then learn the distribution $p(x | y)$ as the authors themselves do in their experiments, for instance rotation, generating a particular color, etc.
> > > > > >
> > > > > > > We trained the model for the same number of iterations as ours
> > > > > >
> > > > > > Did the authors use a pre-trained autoencoder? What model class, were any pre-trained models used, what batch size, the cost of running mmot, how many iterations, etc would be useful to understand the significance and difficulties of the proposed methods.
> > > > > >
> > > > > > Can the authors comment on why they have not done even a MNIST generation experiment. Not being able to scale to ~784 dimensions is not necessarily bad, however it would be useful if the authors can comment on what limitations the method faces when scaling to dimensions > 100.
> > > > > >
> > > > > > My reasons for not increasing my score are:
> > > > > > 1. presentation quality can be improved substantially, both for the methods and the experiments
> > > > > > 2. the authors do not discuss the limitations of their work, particularly scaling when using large batch sizes and dimensions.
> > > > > > 3. the above limitation can be why the authors use low-dimensional problems, limiting to 32 dimensions at most.
> > > > > > 4. the methods section uses the optimal transport plan, at no point is there any discussion of the implications of using finite-sample and mini-batch optimal transport plans on the velocity field they learn.
> > > > > >
> > > > > > Even assuming that they have enough model capacity to learn a vector (or matrix) field, what are the implications of using finite-sample approximations of the transport plan?
> > > > > >
> > > > > > [Chung et al 2022] Diffusion Posterior Sampling for General Noisy Inverse Problems

---

> ### Author Response · Authors · 2024-12-03
> **Reply from authors (1/2)**
>
> Thank you for your response. We would like to clarify our position further.
>
> > So in effect, the authors are not able to upper bound the Dirichlet energy with their objective?
>
> Unbiased estimators of OT plans using finite samples or mini-batches are discussed in the following paper:
>
> - Fatras et al., Learning with minibatch Wasserstein: asymptotic and gradient properties. AISTATS 2020.
>
> It would also be possible to achieve the upper bound of our Dirichlet energy by constructing an estimator in the same manner as they did. In training generative models, we believe in constructing a similar estimator approximately using stochastic gradients.
>
> > Their method is applicable to marginals distributions with continuous-valued support, see definition 1.
>
> Firstly, let us clarify the terminology. In [Definition 1, Albergo et al., 2024], the term "support" refers to the set of data points $x$. In both our paper and Albergo et al.'s paper, $x$ takes continuous values.
> You might be referring to $\Delta^K$ in [Definition 1, Albergo et al., 2024] as the "support." While it is true that $\alpha \in \Delta^K$ takes continuous values, $\alpha$ merely represents a probability vector over a discrete set, making it essentially discrete.
> In contrast, in our setting, the condition vector $c \in \Omega$ not only takes continuous values but can also represent quantities that vary continuously, such as color or angle. This allows for a more flexible representation of continuously varying conditions.
>
> > This is incorrect. In inverse problems, one can define a matrix $ A $ and observations $ y = Ax + \varepsilon $, where $ x $ is the image and $ \epsilon $ is mean zero noise, More typically, one can always consider generating labels $ y = g(x) $ for a deterministic function $ g $.
>
> When considering the relationship $y = Ax + \varepsilon$ for an image $x$ and a categorical label (digit) $y$, if the image $x$ changes continuously, the label $y$ would also change continuously. However, since $y$ is a categorical variable, it is unnatural for it to change continuously.
> In the context of categorical labels, such as digits, the assumption of a continuous relationship does not hold. This is why we believe there is no impartial metric on the conditions of digits when considering a conditional distribution of images, especially because there are no ground-truth datasets other than those used in training (0, ..., 9).
>
> > Did the authors use a pre-trained autoencoder? What model class, were any pre-trained models used, what batch size, the cost of running mmot, how many iterations.
>
> For the image experiment, we used [``Encoder_ResNet_AE_CIFAR``](https://pythae.readthedocs.io/en/latest/models/nn/cifar/resnets.html)
>  of the Pythae library, fine-tuned on the training distributions over 40 epochs with batch size 128.
> The general cost of Sinkhorn MMOT is $O(n^m)$, where $n$ is the batch size and $m$ is the number of marginals.  Indeed, with this scaling, setting $m= |C_{\mathrm{train}}|$ is prohibitive, where $C_{\mathrm{train}}$ is the set of all conditions we use in training.
> We also note that this affects memory availability because storing complexity of $O(n^m)$ is infeasible for large $n$.

---

> ### Author Response · Authors · 2024-12-03
> **Reply from authors (2/2)**
>
> > Can the authors comment on why they have not done even a MNIST generation experiment.
>
> The reason why we did not present the MNIST generation experiment with *digit conditions* is that there is no *continuous* dependence on the digits in the coupling between the conditional distributions.
> The essential purpose of EFM is to generate a *good coupling sample* of $\lbrace x_c \mid c \in \Omega\rbrace$ under the constraint that $x_c$ changes “smoothly” when $c$ is varied over $\Omega$. Recall that the purpose of this was to generate and interpolate/extrapolate the values of $\mu_c$ when $c$ is not in $C_{\mathrm{train}}$. Therefore, we have chosen an experimental setup that has dependencies on $c$.
> If the problem is to generate $x_c \sim  \mu_c$ for each $c$ independently for each $c \in C_{\mathrm{train}}$ (e.g., MNIST with digit conditions), we do not need to consider the *joint* distribution, and it makes no difference whether we use MMOT-EFM or Geo-EFM, or even the EFM with random $\pi$ couplings. In particular, when it comes to generating only the 'marginal distribution' for $c \in C_{\mathrm{train}}$, EFM and [Albergo et al.] are theoretically the same as the original OT-CFM with conditions.
>
> >  it would be useful if the authors can comment on what limitations the method faces when scaling to dimensions > 100.
>
> The effect of *dimension* on the complexity of EFM is indirect. In general, the computational cost of MMOT/OT for $d$ dimensional particles scales linearly with *d*. In contrast, it scales with $n^m$, where $n$ is the number of particles and $m$ is the number of marginal distributions.
> At the same time, the larger the dimension $d$, the slower the *batch* sample converges to the true distribution; in particular,  the expected Wasserstein distance from the empirical distribution to the true distribution scales with $O(n^{-1/d})$ when measured with the $W_1$ distance [3].  Thus, the larger the $d$, the larger the number $n$ of particles it would take to approximate the MMOT/OT with a given precision in terms of the *transport* cost.
>
> [3] Fournier, and Guillin. "On the rate of convergence in Wasserstein distance of the empirical measure." Probability theory and related fields, 2015.
>
>
> > Even assuming that they have enough model capacity to learn a vector (or matrix) field, what are the implications of using finite-sample approximations of the transport plan?
>
> The implications of using finite-sample approximations are twofold.
> (1) The quality of the approximated marginal distributions is affected by the batch size.
> (2) The quality of the approximated coupling (joint distribution) is affected by the batch size.
>
> The first part applies to any batch-based flow-matching method in general because *batches* are used in the flow matching as the approximation of marginal distributions, which converges to the ground-truth distribution. This also applies to the optimal transport cost and, most likely, to Dirichlet energy.
> As stated above, the larger the $d$, the greater the size of the batches it would require to obtain a matrix field with the lowest *transport* energy.
>
> We believe that our response above clarifies the limitations of the method and the impact of mini-batch OT, which are your concerns.

---

### Author Response · Authors · 2024-11-29
**Global comments by authors**

We appreciate the detailed feedback on our paper. Many reviewers pointed out the unclear relationship between smoothness and extrapolation, as well as the scalability issues of MMOT-EFM. In response to these comments, we made the following revisions, which have been reflected in the revised manuscript.

---

## How to Impose Smoothness to Allow Extrapolation to Conditions

First, we would like to clarify the meaning of "smoothness," as mentioned in our motivation. **Our goal is to ensure that the conditional distribution $ p(x \mid c) $ which we will generate is "smooth," meaning it minimizes the Dirichlet energy as defined in Equation (3.2).** Intuitively, the Dirichlet energy represents the sensitivity of the distribution $ p(x \mid c) $ with respect to the conditioning vector $ c $. Specifically, it holds that

$$
\operatorname{Dir}(p)= \text{``}\lim_{\varepsilon\to0}\text{''} C_k \iint_{\Omega \times \Omega}  \frac{W_2^2(p(\cdot \mid c_1), p(\cdot \mid c_2))}{2 \varepsilon^{k+2}} \boldsymbol{1}_{|c_1-c_2| \leqslant \varepsilon} \mathrm{~d} c_1 \mathrm{d} c_2 \quad \text{for} \quad p\colon\Omega\ni c\longmapsto p(\cdot \mid c)\in\mathcal{P}(D),
$$

where $ k $ is the dimension of the condition space $ \Omega $. For the precise meaning of the limit $ \text{``}\lim_{\varepsilon\to0}\text{''} $ and the value of the constant $ C_k $, please refer to [§1.3, Lavenant, 2019].

Thus, **minimizing the Dirichlet energy implies that the sensitivity with respect to the condition $ c $ is not too large.**

Although there are infinite ways of extrapolation, it is reasonable to assume an inductive bias that the sensitivity of data in nature (e.g., molecules) to conditions (e.g., chemical properties) is not unnaturally large. Therefore, our method addresses extrapolation by learning a model such that the data to be extrapolated follows this inductive bias of low sensitivity. We would like to note that this kind of inductive bias has been used throughout the history of generative models as a method to prevent overfitting and stabilize generative models; see, for example, Miyato et al. in ICLR, 2018.

Our experiments in §7 demonstrate that EFM, which minimizes the Dirichlet energy, outperforms methods that do not minimize this energy (such as FM and COT-FM) in terms of generation performance.

In addition, the cost (objective) function used in our multi-marginal optimal transport (MMOT) approach provides an upper bound on the Dirichlet energy; please refer to lines 233-236 and Table 1. Therefore, optimizing the transport plan $ \pi $ through the MMOT approach also minimizes the Dirichlet energy, which in turn reduces the sensitivity of the generated distribution $ p(x \mid c) $ with respect to the conditioning vector $ c $.

---

## Scalability of EFMs

Because we mentioned the complexity of MMOT in the manuscript, many reviewers were concerned about the scalability of the proposed method. **The new experiment we conducted in §7.2 to generate images with continuous conditions is expected to dispel this concern.**

In §7.2 of the revised manuscript, we included the conditional generation of the MNIST dataset with a single ImageNet background, where the two conditions are "color" and "rotation," which form a 4-dimensional condition space $ \Omega=[0, 1]^4 $ (the first dimension is rotation, and the rest are RGB). For training, we used a conditional dataset corresponding to 12 uniform random samples of $ \Omega $. We evaluated the Wasserstein distance from the ground truth (GT) conditional distributions in a manner similar to Figure 4. Note that these conditional distributions are nonlinear in the presence of background. We compared our method with the classifier-free guidance method (Zheng et al. [1]) with different guidance strengths. On this dataset, we see that the Wasserstein error increases monotonically with the distance of the target conditions from the training set of conditions (for example, $ c=[0, 0, 0, 0] $ is very far from the training conditions, and it is much more difficult to realize).

Please note that we had to make color and rotation the choice of conditions in this experiment because the default conditional labels of the image benchmarks, such as MNIST/CIFAR10, are (1) discrete, and (2) there is no impartial metric on the space of label conditions. For example, MNIST is labeled with digits from 0 to 9, but in terms of image generation, 0 is no closer to 1 than it is to 9. The one-hot vector embedding is also not very reflective of actual image generation because 9 is actually much closer to 4 in shape than it is to 3. This is the reason why we limited our experiments to datasets like ZINC-250k of the form $ (x, c(x)) $, where $ c(x) $ is a feature of $ x $.

---

The above explanations have been added in purple highlights in the revised manuscript. We hope that these sections will make the novelty and potential of our research clearer.

---

### Meta-Review · Area_Chair_piLa · 2024-12-19

**Metareview:**

The paper proposes a new flow matching algorithm based on a generalized (matrix-valued) continuity equation, with the motivation being unobserved data/conditions. The main claim of the work is that it outperforms existing methods in molecular generation tasks with sparsely observed conditions.

The main strength is the rigorous development of the proposed algorithm, and clear motivation of the problem. However, the main claim of the paper that the method outperforms existing approaches is not fully validated in the experiments, and it remains unclear whether there is practical setting in which the generalized continuity equation is useful.

I recommend to reject the paper at this stage, and encourage the authors to take the reviewers details feedback into account for a resubmission.

**Additional Comments On Reviewer Discussion:**

- Reviewer a59p pointed out that there is not really a practical setting in which the method is useful
- Reviewer QpJX pointed out various issues with the theory.

Both were not cleared after discussions in the rebuttal phase, which influenced my decision to reject the paper.

---

### Decision · Program_Chairs · 2025-01-22

Reject